# Contrastively Disentangled Sequential Variational Autoencoder

**Junwen Bai**
Cornell University
jb2467@cornell.edu

**Weiran Wang**
Google
weiranwang@google.com

**Carla Gomes**
Cornell University
gomes@cs.cornell.edu

## Abstract

Self-supervised disentangled representation learning is a critical task in sequence modeling. The learnt representations contribute to better model interpretability as well as the data generation, and improve the sample efficiency for downstream tasks. We propose a novel sequence representation learning method, named Contrastively Disentangled Sequential Variational Autoencoder (C-DSVAE), to extract and separate the static (time-invariant) and dynamic (time-variant) factors in the latent space. Different from previous sequential variational autoencoder methods, we use a novel evidence lower bound which maximizes the mutual information between the input and the latent factors, while penalizes the mutual information between the static and dynamic factors. We leverage contrastive estimations of the mutual information terms in training, together with simple yet effective augmentation techniques, to introduce additional inductive biases. Our experiments show that C-DSVAE significantly outperforms the previous state-of-the-art methods on multiple metrics.

## 1   Introduction

The goal of self-supervised learning methods is to extract useful and general representations without any supervision, and to further facilitate downstream tasks such as generation and prediction [1]. Despite the difficulty of this task, many existing works have shed light on this field across different domains such as computer vision [2, 3, 4, 5], natural language processing [6, 7, 8] and speech processing [9, 10, 11, 12, 13] (also see a huge number of references in these papers). While the quality of the learnt representations improves gradually, recent research starts to put more emphasis on learning disentangled representations. This is because disentangled latent variables may capture separate variations of the data generation process, which could contain semantic meanings, provide the opportunity to remove unwanted variations for a lower sample complexity of downstream learning [14, 15], and allow more controllable generations [16, 17, 18]. These advantages lead to a rapidly growing research area, studying various principles and algorithmic techniques for disentangled representation learning [19, 20, 21, 22, 23, 24, 25, 26]. One concern raised in [24] is that without any inductive bias, it would be extremely hard to learn meaningful disentangled representations. On the other hand, this concern could be much alleviated in the scenarios where the known structure of the data can be exploited.

In this work, we are concerned with the representation learning for sequence data, which has a unique structure to utilize for disentanglement learning. More specifically, for many sequence data, the variations can be explained by a dichotomy of a static (time-invariant) factor and dynamic (time-variant) factors, each varies independently from the other. For example, representations of a video recording the movements of a cartoon character could be disentangled into the character identity (static) and the actions (dynamic). For audio data, the representations shall be able to separate the speaker information (static) from the linguistic information (dynamic).

35th Conference on Neural Information Processing Systems (NeurIPS 2021).

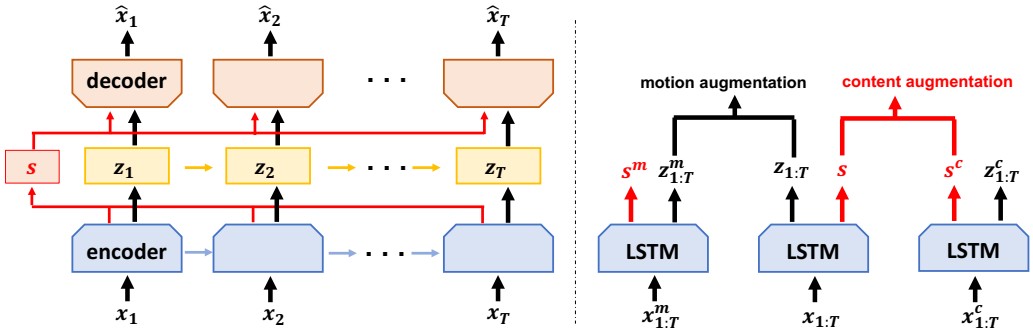

Figure 1: The illustration of our C-DSVAE model. **Left panel** is the general structure of the sequence-to-sequence auto-encoding process: each frame is passed to the LSTM cell; dynamic factors $z_{1:T}$ are extracted for each time step; the static factor $s$ is extracted by summarizing the full sequence; the generation/reconstruction of frame $i$ depends on $s$ and $z_i$. **Right panel** depicts the contrastive learning module of C-DSVAE: $x_{1:T}^m$ is the motion augmentation of $x_{1:T}$ and $x_{1:T}^c$ is the content augmentation of $x_{1:T}$; dynamic factors $z_{1:T}^m$ of $x_{1:T}^m$ can be seen as the positive sample for the anchor $z_{1:T}$ in the contrastive estimation w.r.t. the motion, and similarly the static factor $s^c$ of $x_{1:T}^c$ can be viewed as the positive sample for $s$ w.r.t. the content.

We propose Contrastively Disentangled Sequential Variational Autoencoder (C-DSVAE), a method seeking for a clean separation of the static and dynamic factors for the sequence data. Our method extends the previously proposed sequential variational autoencoder (VAE) framework, and performs learning with a different evidence lower bound (ELBO) which naturally contains mutual information (MI) terms to encourage disentanglement. Due to the difficulty in estimating high dimensional complex distributions (e.g., for the dynamic factors), we further incorporate the contrastive estimation for the MI terms with systematic data augmentation techniques which modify either the static or dynamic factors of the input sequence. The new estimation method turns out to be more effective than the minibatch sampling based estimate, and introduces additional inductive biases towards the invariance. To our knowledge, we are the first to synergistically combine the contrastive estimation and sequential generative models in a principled manner for learning disentangled representations. We validate C-DSVAE on four datasets from the video and audio domains. The experimental results show that our method consistently outperforms the previous state-of-the-art (SOTA) methods, both quantitatively and qualitatively.

## 2 Method

We denote the observed input sequence as $x_{1:T} = \{x_1, x_2, ..., x_T\}$ where $x_i$ represents the input feature at time step $i$ (e.g., $x_i$ could be a video frame or the spectrogram feature of a short audio segment), and $T$ is the sequence length. The latent representations are divided into the static factor $s$, and the dynamic factors $z_{1:T}$ where $z_i$ is the learnt dynamic representation at time step $i$.

### 2.1 Probabilistic Model

We assume that in the ground-truth generation process, $z_i$ depends on $z_{<i} = \{z_0, z_1, ..., z_{i-1}\}$ where $z_0 = \mathbf{0}$, and the observation $x_i$ is independent of other frames conditioned on $z_i$ and $s$. Furthermore, we assume the static variable $s$ and the dynamic variables $z_{1:T}$ are independent from each other, i.e., $p(s, z_{1:T}) = p(s)p(z_{1:T})$. Formally, let $z = (s, z_{1:T})$ and we have the following complete likelihood

$$p(x_{1:T}, z) = p(z)p(x_{1:T}|z) = \left[p(s)\prod_{i=1}^{T} p(z_i|z_{<i})\right] \cdot \prod_{i=1}^{T} p(x_i|z_i, s) \qquad (1)$$

where $p(z)$ is the prior for sampling $z$. Our formulation captures the general intuition that we can separate the variations of the sequence into the time-dependent dynamic component (described by $z_i$'s), and a static component (described by $s$) which remains the same for different time steps in the same sequence but differs between sequences. For example, in the cartoon character video

(see Figure 3), the hair, shirt and pants should be kept the same when the character walks around, but different videos can have different characters. Similarly, for a speech utterance, the phonetic transcription controls the vocal tract motion and the sound being produced over time, but the speaker identity remains the same for the whole utterance. In this work, we refer to the dynamic component as "motion" and the static component as "content".

For the prior distributions, we choose $p(s)$ to be the standard Gaussian $\mathcal{N}(0, I)$, and $p(z_i|z_{<i})$ to be $\mathcal{N}(\mu(z_{<i}), \sigma^2(z_{<i}))$ where $\mu(\cdot)$ and $\sigma(\cdot)$ are modeled by LSTMs [27]. In operations, the drawn sample $s$ is shared throughout the sequence. To draw a sample of $z_t$, we first take a sample of $z_{t-1}$ as the input of the current LSTM cell. Then forwarding the LSTM for one step gives us the distribution of $z_t$, from which we draw a sample with the reparameterization trick [28].

To extract the latent representations given only the observed data $x_{1:T}$ where the motion and content are mixed together, we hope to learn a posterior distribution $q(z|x_{1:T})$ where the two components are disentangled. That is, similar to the prior, our posterior should have a factorized form:

$$q(z|x_{1:T}) = q(z_{1:T}, s|x_{1:T}) = q(z_{1:T}|x_{1:T})q(s|x_{1:T}) = q(s|x_{1:T}) \prod_{i=1}^{T} q(z_i|z_{<i}, x_{\leq i}). \qquad (2)$$

The posterior distributions are also modeled by LSTMs. A nested sampling procedure, resembling that of the prior, is applied to the posterior of dynamic variables. Similar parameterizations of dynamic variables by recurrent networks have been proposed in prior works [29, 30, 31]. The standard loss function for learning the latent representations is an ELBO [32, 33] (also see a derivation in Appendix A.1):

$$\max_{p,q} \mathbb{E}_{x_{1:T} \sim p_D} \mathbb{E}_{q(z|x_{1:T})} \left[ \log p(x_{1:T}|z) - KL[q(z|x_{1:T})||p(z)] \right] \qquad (3)$$

where $p_D$ is the empirical data distribution. Under the parameterization of the posterior where $s$ and $z_{1:T}$ are mutually independent, the KL-divergence term reduces to

$$KL[q(z|x_{1:T})||p(z)] = KL[q(s|x_{1:T})||p(s)] + KL[q(z_{1:T}|x_{1:T})||p(z_{1:T})] \qquad (4)$$

where the second term is approximated with the sampled trajectories of the dynamic variables $z_{1:T}$.

## 2.2 Our approach: Mutual information-based disentanglement

Several existing sequence representation learning methods [32, 33, 18] are built on top of the loss function (3). However, this formulation also brings several issues. The KL-divergence regularizes the posterior of the static or dynamic factors to be close to the corresponding priors. When modeling them with powerful neural architectures like LSTMs, it is possible for the KL-divergence to be close to zero, yet at the same time, the posteriors become non-informative of the inputs; this is a common issue for deep generative models like VAE [34]. Techniques have been proposed to alleviate this issue, including adjusting the relative weights of the loss terms to regularize the capacity of posteriors [20], replacing the individual posteriors in the KL terms with aggregated posteriors [35, 36, 37] and enforcing latent structures (such as disentangled representations) in the posteriors [23, 22, 38].

Our principled approach for learning useful representations from sequences is inspired by these prior works, and at the same time incorporates the unique sequential structure of the data. Without inductive biases, the goal of disentanglement can hardly be achieved since it is possible to find entangled $s$ and $z_{1:T}$ that explain the data equally well (in fact, by Theorem 1 of [24], there could exist infinitely many such entangled factors). The problem may seem less severe in our setup, since $s$ is shared across time and it is hard for such $s$ to capture all dynamics. Nevertheless, since both $s$ and $z_i$ are used in generating $x_i$, it is still possible for $z_i$ to carry some static information. In the extreme case where each $z_i$ encompasses $s$, $s$ would no longer be indispensable for the generation. This issue motivated prior works to optimize the (estimate of) mutual information among latent variables and inputs [33, 18, 39].

Our method seeks to achieve clean disentanglement of $s$ and $z_{1:T}$, by optimizing the following objective function, which introduces additional MI terms to the vanilla ELBO in (3):

$$\max_{p,q} \mathbb{E}_{x_{1:T} \sim p_D} \mathbb{E}_{q(z|x_{1:T})}[\log p(x_{1:T}|z)] - KL[q(z)||p(z)]$$
$$= \mathbb{E}_{x_{1:T} \sim p_D} \mathbb{E}_{q(z|x_{1:T})}[\log p(x_{1:T}|z)] - (KL[q(s|x_{1:T})||p(s)] + KL[q(z_{1:T}|x_{1:T})||p(z_{1:T})])$$
$$+ I_q(s; x_{1:T}) + I_q(z_{1:T}; x_{1:T}) - I_q(z_{1:T}; s) \qquad (5)$$

where the aggregated posteriors are defined as $q(z) = q(s, z_{1:T}) = \mathbb{E}_{p_D}[q(s|x_{1:T})q(z_{1:T}|x_{1:T})]$, $q(s) = \mathbb{E}_{p_D}[q(s|x_{1:T})]$, $q(z_{1:T}) = \mathbb{E}_{p_D}[q(z_{1:T}|x_{1:T})]$, and the MI terms are defined as $I_q(s; x_{1:T}) = \mathbb{E}_{q(s,x_{1:T})}\left[\log \frac{q(s|x_{1:T})}{q(s)}\right]$, $I_q(z_{1:T}; x_{1:T}) = \mathbb{E}_{q(z_{1:T},x_{1:T})}\left[\log \frac{q(z_{1:T}|x_{1:T})}{q(z_{1:T})}\right]$ (and $I_q(z_{1:T}; s)$ is defined similarly). The intuition behind (5) is simple: besides explaining the data and matching the posteriors with priors, $z_{1:T}$ and $s$ shall contain useful information from $x_{1:T}$ while excluding the redundant information from each other. We further justify (5) by showing that it still forms a valid ELBO.

**Theorem 1.** *With our parameterization of $(s, z_{1:T})$, (5) is a valid lower bound of the data log-likelihood $\mathbb{E}_{x_{1:T} \sim p_D} \log(x_{1:T})$.*

The full proof can be found in Appendix A.2. With this guarantee, we can follow the spirit of [20, 23] to add and adjust the additional weight coefficients $\alpha$ to the KL terms, $\beta$ to the MI terms $I_q(s; x_{1:T})$, $I_q(z_{1:T}; x_{1:T})$, and $\gamma$ to $I_q(z_{1:T}; s)$,

$$\mathbb{E}_{x_{1:T} \sim p_D} \mathbb{E}_{q(z|x_{1:T})}[\log p(x_{1:T}|z)] - \alpha(KL[q(s|x_{1:T})||p(s)] + KL[q(z_{1:T}|x_{1:T})||p(z_{1:T})]) \\ + \beta(I_q(s; x_{1:T}) + I_q(z_{1:T}; x_{1:T})) - \gamma I_q(z_{1:T}; s). \tag{6}$$

It remains to estimate the objective (6) for optimization. The KL term for $z_{1:T}$ is estimated with the standard Monte-Carlo sampling, using trajectories of $z_{1:T}$ [32]. The KL term for $s$ can be estimated analytically. For the MI terms, we attempt two estimations. The first estimation uses a standard mini-batch weighted sampling (MWS) following [23, 33]. Due to the high dimensionality of $z_{1:T}$ and the complex dependency among time steps, it may be hard for MWS to estimate the distributions accurately, so we also explore non-parametric contrastive estimations for $I_q(s; x_{1:T})$ and $I_q(z_{1:T}; x_{1:T})$ with additional data augmentations, which we will detail below.

### 2.3 C-DSVAE: Contrastive estimation with augmentation

A contrastive estimation of $I(z_{1:T}; x_{1:T})$ can be defined as follows

$$\mathcal{C}(z_{1:T}) = \mathbb{E}_{p_D} \log \frac{\phi(z_{1:T}, x_{1:T}^+)}{\phi(z_{1:T}, x_{1:T}^+) + \sum_{j=1}^n \phi(z_{1:T}, x_{1:T}^j)} + \log(n+1) \tag{7}$$

where $x^+$ is a "positive" sequence, while $x^j$, $j = 1, \ldots, n$ is a collection of $n$ "negative" sequences. When $\phi(z_{1:T}, x_{1:T}) = \frac{q(x_{1:T}|z_{1:T})}{q(x_{1:T})}$, one can show that (7) approximates $I_q(z_{1:T}, x_{1:T})$; see Appendix A.4 for a proof. To implement (7), $z_{1:T}$ is the trajectory obtained by using the mean at each time step from $q(z_{1:T}|x_{1:T})$ for a input sequence $x_{1:T}$, while $x_{1:T}^j$ is a randomly sampled sequence from the minibatch. A similar contrastive estimation is defined for $s$ as well. To provide meaningful positive sequences that encourage the invariance of the learnt representations, we obtain $x_{1:T}^+$ by systematically perturbing $x_{1:T}$. In Sec A.4, we discuss how the augmented data give good estimate of (7) under additional assumptions.

**Content augmentation** The static factor (e.g., the character identity in videos or the speaker in audios) is shared across all the time steps, and should not be affected by the exact order of the frames. We therefore randomly shuffle or simply reverse the order of time steps to generate the content augmentation of $x_{1:T}$ and denote it by $x_{1:T}^c$. The static and dynamic latent factors of $x_{1:T}^c$, modeled by $q$, are denoted by $s^c$ and $z_{1:T}^c$. This is an inexpensive yet useful strategy, applicable to both audios and videos. Similar ideas were introduced in [33] albeit not used for contrastive estimations.

**Motion augmentation** In motion augmentation, we would like to maintain the dynamic factors (e.g., actions or movements) while replacing the content with a meaningful alternative. Thanks to the recent efforts in contrastive learning, multiple effective strategies have been proposed. For video datasets, we adopt the combination of cropping, color distortion, Gaussian blur and reshaping [4, 40]. For audio datasets, we use classical unsupervised voice conversion algorithms [41, 42]. The motion augmentation of $x_{1:T}$ is denoted by $x_{1:T}^m$, with the static factor $s^m$ and dynamic factors $z_{1:T}^m$ estimated by $q$.

Note that $x_{1:T}^m$ is the motion augmentation of $x_{1:T}$, and $x_{1:T}$ in turn is also the motion augmentation of $x_{1:T}^m$. Similarly, $s$ and $s^c$ are mutually the "positive" sample to each other w.r.t. the static factor. During the training, it is efficient to generate the augmentations for each sequence in the minibatch,

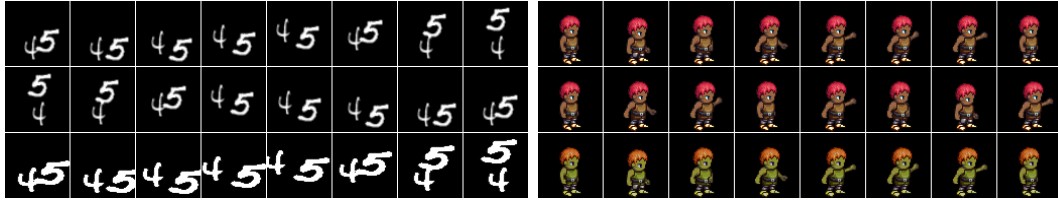

Figure 2: Data augmentations on SM-MNIST and Sprites. **Left panel**: SM-MNIST. The first row is the raw input sequence of moving digits. The second row reverses the sequence order of the raw sequence, serving as the content augmentation. The third row stretches the frames and enhances the color, which changes the digit styles but does not change the digit movements and thus forms a motion augmentation. **Right panel**: Sprites. The first row is the raw input character video. The second row is the content augmentation with a random order. The third row is a motion augmentation produced by distorting colors and adding Gaussian noise.

and apply contrastive estimation on both the original data and the augmented data. This leads to the following final estimates:

$$I_q(z_{1:T}; x_{1:T}) \approx \frac{1}{2}(\mathcal{C}(z_{1:T}) + \mathcal{C}(z_{1:T}^m)), \qquad I_q(s; x_{1:T}) \approx \frac{1}{2}(\mathcal{C}(s) + \mathcal{C}(s^c)) \qquad (8)$$

where we use $\phi(z_{1:T}(x_{1:T}), x_{1:T}^*) = \exp(\text{sim}(z_{1:T}, z_{1:T}^*)/\tau)$ with $^*$ indicating the latent variables for the positive/augmented or negative sequences (extracted from $q$). $\text{sim}(\cdot, \cdot)$ is the cosine similarity function and $\tau = 0.5$ is a temperature parameter. This form of $\phi$ using cosine similarity is widely adopted in contrastive learning [4, 43] as it removes one degree of freedom (length) for high dimensional feature space. Plugging (8) into (6) gives our final learning objective with contrastive estimation. We name our method Contrastively Disentangled Sequential Variational Encoder (C-DSVAE), and a full model illustration is shown in Figure 1.

In practice, we find the contrastive estimation is more effective than the MWS based on Gaussian probabilities (see an ablation study in Appendix F). Interestingly, we observe that with only the contrastive estimation and augmentations, $I_q(z_{1:T}; s)$ (last term in (5)) decreases even if we do not include its estimation in our optimization process (see Appendix E), indicating that a good disentanglement between $z_{1:T}$ and $s$ could be attained through contrastive estimations solely.

As it will become evident in the experiments, our method achieves a cleaner separation of the dynamic and static factors than previous methods. We believe this is partly due to the inductive bias introduced by our approach: in order to consistently map two sequences with the same motion but different contents (which vary independently from the motion according to the generation process (1)) to the same latent representation $z_{1:T}$, we must discard the information of the inputs regarding $s$; similar arguments hold for the static variables. At the same time, we enforce that the extracted $z_{1:T}$ and $s$ together should reconstruct $x_{1:T}$ so that no information is lost in the auto-encoding process. To our knowledge, we are the first to use the contrastive estimation with augmentations in the sequential VAE scenario, for pushing the disentanglement of variations.

## 3 Related Work

In terms of the general frameworks, disentangled representation learning methods can be categorized into GAN-based [44] and VAE-based [28] approaches. GAN-based methods like MoCoGAN [45] aim at generating videos from content noise and motion noise. These models typically do not explicitly learn the encoder from inputs to the latent variables, rendering them less applicable for representation learning.

The VAE framework parameterizes both the encoder (variational posterior) and the decoder (reconstruction), and optimize them jointly with a well-defined ELBO objective. The encoder allows us to directly control and reason about the properties of the extracted features, which can be straightforwardly used for downstream tasks. Many variants of VAE were proposed to encourage the disentanglement/interpretability of *individual latent dimensions*. $\beta$-VAE [20] adds a coefficient to the per-sample KL-divergence term to better constrain the information bottleneck. FactorVAE [22] and

$\beta$-TCVAE [23] explicitly separate out the total correlation term, and adjust its weight coefficient in the objective to encourage the per-dimensional disentanglement of the *aggregated* posterior. From a theoretical standpoint, [24] points out that without further inductive biases, it is impossible to learn disentangled representations with the standard VAE.

To handle the sequence data, vanilla VAE is extended to the recurrent version, and prior works have explored different approaches for separating the content and the motion. FHVAE [46] designs a hierarchical VAE model which uses "sequence-dependent variables" (corresponding to the speaker factor) and "sequence-independent variables" (corresponding to the linguistic factors) for modeling speech sequences, and performs learning with the per-sequence ELBO while respecting the hierarchy in designing the posterior. FHVAE does not use additional loss terms for encouraging the disentanglement. Different from FHVAE, DSVAE [32] explicitly models the static and dynamic factors in its graphical model and its posterior has a factorized form. A few recent works (including ours) inherit the clean formulation of DSVAE in terms of the graphical model and prior/posterior parameterization. S3VAE [33] introduces additional loss terms to the objective of DSVAE in an ad-hoc fashion, such as a triplet loss that encourages the invariance of the learnt static factors by permuting the frame order (spiritually similar to our contrastive estimation for $I(s, x_{1:T})$), an additional prediction loss on $z_{1:T}$ leveraging external supervision for the motion labels, and an MI term of $I(s, z_{1:T})$ to be minimized. Our C-DSVAE differs from S3VAE in that we naturally introduce the MI terms from the fundamental principle of VAE, and use augmentations instead of external supervision for learning the static and dynamic factors. Also based on the DSVAE formulation, R-WAE [18] proposes to replace the distance measure between the aggregated posterior and the prior with the Wasserstein distance, in the belief that the KL divergence is too restrictive. In R-WAE, Maximum Mean Discrepancy (MMD) based estimation or GAN-based estimation of Wasserstein distance is used, either of which requires heavy tuning of hyperparameters. Another similar method, IDEL [47], optimizes the additional set of MI terms similar to ours. However, in our work, we show that these MI terms can be naturally derived from the fundamental principle of VAE, and we additionally propose the contrastive estimation and data augmentations to strengthen them, which was not done before.

Another relevant research area is nonlinear ICA, which tries to understand the conditions under which the disentanglement can be achieved, possibly with contrastive learning or VAE [48, 49, 50, 25, 26]. Our general approach is well-aligned with the current understanding in this direction. That is, while general disentanglement (per-dimensional disentanglement) is hard to achieve without stringent assumptions on the data generation process, we can seek some certain level of disentanglement (in our case, the group-wise disentanglement between the static and dynamic factors) given auxiliary information (which we provide through augmentations). Our intuitive arguments for the reason of separation (see the last paragraph of Sec 2.3) show that VAE and contrastive learning may be complementary to each other, and combining them could potentially gain better disentanglement.

# 4 Experiments

We compare C-DSVAE with the state-of-the-art sequence disentanglement learning methods: FHVAE [46], MoCoGAN [45], DSVAE [32], S3VAE [33] and R-WAE [18], with the same experimental setups used by them.

## 4.1 Datasets

**Sprites** [51] is a cartoon character video dataset. Each character's (dynamic) motion can be categorized into three actions (walking, spellcasting, slashing) and three directions (left, front, right). The (static) content comprises of each character's skin color, tops color, pants color and hair color. Each color has six variants. Every sequence is composed of 8 frames of RGB images with size $64 \times 64$.

**MUG** [52] is a facial expression video dataset. The static factor corresponds to a single person's identity. Each individual performs six expressions (motion): anger, fear, disgust, happiness, sadness and surprise. Following [45], each sequence contains 15 frames of RGB images with size $64 \times 64$ (after resizing).

**SM-MNIST** ([53], Stochastic Moving MNIST) is a dataset that records the random movements of two digits. Each sequence contains 15 gray scale images of size $64 \times 64$. Individual digits are collected from MNIST.

Table 1: Disentanglement metrics on Sprites.

| Methods | Acc↑ | IS↑ | H(y\|x)↓ | H(y)↑ |
|---|---|---|---|---|
| MoCoGAN | 92.89% | 8.461 | 0.090 | 2.192 |
| DSVAE | 90.73% | 8.384 | 0.072 | 2.192 |
| R-WAE | 98.98% | 8.516 | 0.055 | 2.197 |
| S3VAE | 99.49% | 8.637 | 0.041 | 2.197 |
| C-DSVAE | **99.99%** | **8.871** | **0.014** | **2.197** |

Table 2: Disentanglement metrics on MUG.

| Methods | Acc↑ | IS↑ | H(y\|x)↓ | H(y)↑ |
|---|---|---|---|---|
| MoCoGAN | 63.12% | 4.332 | 0.183 | 1.721 |
| DSVAE | 54.29% | 3.608 | 0.374 | 1.657 |
| R-WAE | 71.25% | 5.149 | 0.131 | 1.771 |
| S3VAE | 70.51% | 5.136 | 0.135 | 1.760 |
| C-DSVAE | **81.16%** | **5.341** | **0.092** | **1.775** |

Table 3: Disentanglement metrics on SM-MNIST.

| Methods | Acc↑ | IS↑ | H(y\|x)↓ | H(y)↑ |
|---|---|---|---|---|
| MoCoGAN | 74.55% | 4.078 | 0.194 | 0.191 |
| DSVAE | 88.19% | 6.210 | 0.185 | 2.011 |
| R-WAE | 94.65% | 6.940 | 0.163 | 2.147 |
| S3VAE | 95.09% | 7.072 | 0.150 | 2.106 |
| C-DSVAE | **97.84%** | **7.163** | **0.145** | **2.176** |

Table 4: Disentanglement metrics on TIMIT.

| Methods | content EER↓ | motion EER↑ |
|---|---|---|
| FHVAE | 5.06% | 22.77% |
| DSVAE | 5.64% | 19.20% |
| R-WAE | 4.73% | 23.41% |
| S3VAE | 5.02% | 25.51% |
| C-DSVAE | **4.03%** | **31.81%** |

**TIMIT** [54] is a corpus of read speech for acoustic-phonetic studies and speech recognition. The utterances are produced by American speakers of eight major dialects reading phonetically rich sentences. Following [46, 18], we extract per-frame spectrogram features (with a shift size of 10ms) from audio, and segments of 200ms duration (20 frames) are chunked from the original utterances and then treated as independent sequences for learning. All datasets are separated into training, validation and testing splits following [46, 33, 18].

## 4.2 Disentanglement Metrics

The quantitative performance measures all the models from two aspects: first, by fixing either the static or dynamic factors and randomly sample the other, how well the fixed factor can be recognized; and second, with the randomly sampled factor, how different the generated sequence is from the original one. To this end, we pretrain a separate classifier $C$ to identify the static or dynamic factors (if available). The classifier $C$ is carefully trained with the full supervision and thus qualifies as a judge.

We use five metrics to evaluate the disentanglement performance: accuracy, Inception Score, inter-entropy, intra-entropy and equal error rate. Accuracy (Acc) measures how well the fixed factor can be identified by the classifier $C$. Inception Score ($IS$) [55] computes the KL-divergence between the conditional predicted label distribution $p(y|x_{1:T})$ and the marginal predicted label distribution $p(y)$ from $C$. Inter-Entropy is similar to IS, but measures the diversity solely on the marginal predicted label distribution $p(y)$; a higher inter-entropy indicates that the generated sequences are more diverse. Intra-Entropy measures the entropy over $p(y|x)$; a lower intra-entropy means more confident predictions. Equal error rate (EER) [56] is used in the TIMIT experiment to evaluate the speaker identification accuracy.

## 4.3 Hyperparameters and Architectures

As is commonly done in VAE learning [20], in (6) we add coefficients $\alpha$ to the KL terms, $\beta$ to the contrastive terms, while the coefficient $\gamma$ of $I(s; z_{1:T})$ is fixed to be 1. In our experiments, $\alpha$ is tuned over $\{0.6, 0.9, 1.0, 2.0\}$ and $\beta$ is tuned over $\{0.1, 0.2, 0.5, 0.7, 1.0, 2.0, 5.0\}$. We use the Adam optimizer [57] with the learning rate chosen from $\{0.0005, 0.001, 0.0015, 0.002\}$ through grid search. Models are trained until convergence and the one with the best validation accuracy would be chosen for testing. Our network architecture follows [33]: motion priors and posteriors are both parameterized by uni-directional LSTMs, which output $\mu$ and $\sigma$ at each time step for the dynamic factors, while the content posterior produces $\mu$ and $\sigma$ for the static factor of the whole sequence with a bi-directional LSTM. See Appendix C.2 for further details.

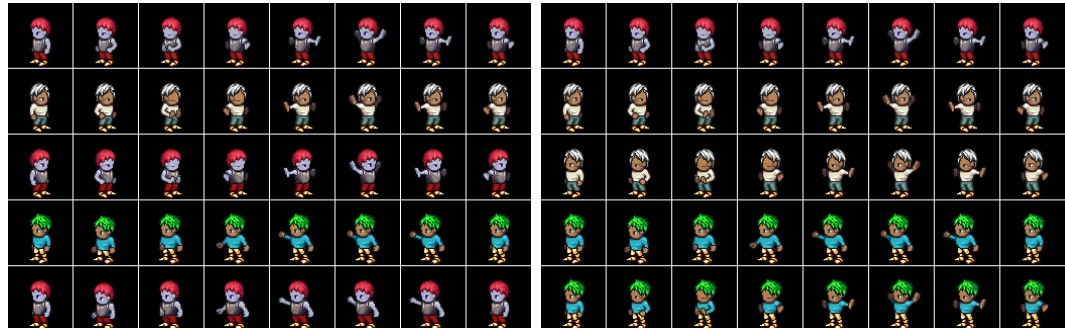

Figure 3: Manipulate the static and dynamic factors on Sprites. Row 1, 2, 4 are the raw test input sequences, while row 3, 5 are manipulated generations. **Left panel**: Row 3 uses $s$ from row 1 and $z_{1:T}$ from row 2. Row 5 uses $s$ from row 1 and $z_{1:T}$ from row 4. **Right panel**: Row 3 uses $z_{1:T}$ from row 1 and $s$ from row 2. Row 5 uses $z_{1:T}$ from row 1 and $s$ from row 4.

## 4.4 Quantitative Results

Tables 1, 2, 3, 4 demonstrate the disentanglement evaluations on the datasets; ↑ means the higher the value the better the performance, and ↓ means the reverse. In Table 1, we fix $z_{1:T}$ and randomly sample $s$ from $p(s)$ for the evaluation. We observe that R-WAE, S3VAE, C-DSVAE can all deliver high accuracies in predicting a total of 9 classes (3 actions × 3 directions), but C-DSVAE outperforms all the others. On the other hand, when we fix $s$ and randomly sample $z_{1:T}$, all the methods achieve near-perfect accuracies for predicting character identities (and thus we do not show this result here). Similarly, in Table 2, we evaluate the ability to maintain the facial expression by sampling $s$ and fixing $z_{1:T}$. C-DSVAE outperforms R-WAE and S3VAE by over 10% relatively in the prediction accuracy, and its intra-entropy is lower by 30% relatively. Table 3 measures how well the static factor $s$, which corresponds to one of the total 100 digit combinations, can be recognized when the dynamic factors are randomly sampled (the dynamic factors are continuous and thus hard to be categorized). While R-WAE, S3VAE, and C-DSVAE can all generate high-quality moving digits (see Sec 4.5), C-DSVAE non-trivially outperforms the others on all metrics.

For TIMIT, we extract both the content factor ($s$, corresponds to the speaker identity) and the dynamic factors ($z_{1:T}$, correspond to the linguistic content) for the sequences. And for each of them, we compute the cosine similarity of representations between pairs of sequences, based on which we perform thresholding to classify if the two sequences are produced by the same speaker. Through varying the threshold, we compute the EER for the speaker verification task [46, 32]. For a well disentangled latent space, we expect the EER on $s$ (content EER) to be low, while the EER on $z_{1:T}$ (motion EER) to be high. The EER results are given in Table 4. Compared with the previous SOTA by R-WAE, C-DSVAE reduces the content EER by over 10% relatively.

## 4.5 Qualitative Results

We now compare different methods qualitatively with the following tasks: given the extracted static factor $s$ and dynamic factors $z_{1:T}$, we fix one of them and manipulate the other to see if we can observe the desired variations in the generated sequences. Figure 3 gives example generations for Sprites. We observe that the character identities are well preserved when we fix $s$ (left) and the motion is well maintained when we fix $z_{1:T}$ (right), demonstrating the clean disentanglement by C-DSVAE. Figure 4 compares the generated sequences when the motion (facial expression) is fixed for C-DSVAE and S3VAE on MUG. We replace $z_{1:T}$ for each input sequence with the one from the first row, and therefore the generated sequences are expected to have the smiling expression. C-DSVAE and S3VAE both generate recognizable smiling faces and preserve the individual's identity, but the sequences generated by S3VAE have blurs and are less sharp. In Figure 5, we fix the digit movements and take the content (digit identities) from various other sequences. Entangled factors might cause the extracted $z_{1:T}$ to carry information of $s$. Row 3 of Figure 5b shows such an example produced by R-WAE where the motion factor of one digit carries the content information of "9" from row 1.

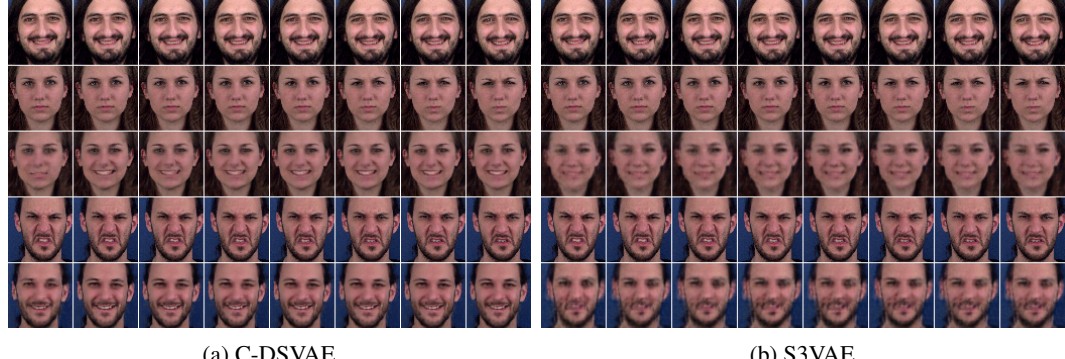

|(a) C-DSVAE|(b) S3VAE|

Figure 4: Fix the facial expression on MUG. Row 1, 2, 4 are the raw test sequences with different facial expressions. We replace $z_{1:T}$ of row 2, 4 with $z_{1:T}$ of row 1 and compare the performance of C-DSVAE and S3VAE. Hence row 3, 5 are expected to display happiness as well. Our C-DSVAE's generations are clean and sharp, while S3VAE produces blurred sequences (zoom in to see the details).

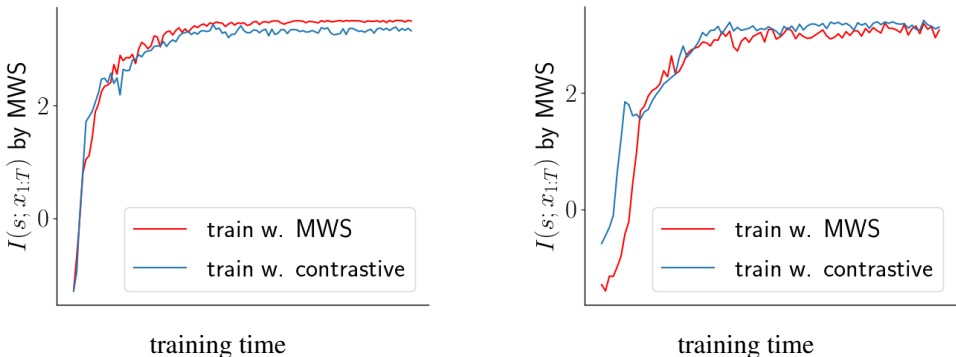

|(a) C-DSVAE|(b) R-WAE|

Figure 5: Fix the movement and replace the digits. Row 1, 2, 4 are input test sequences with different digits and movements. Row 3, 5 use $z_{1:T}$ from row 1 while retaining their own $s$. The sequences generated by C-DSVAE are clean and consistent, while R-WAE sometimes produces blurred or incorrect digits.

Figure 6: Learning curves of the MI terms (estimated by MWS) between latent factors and the input sequences, when the training uses either MWS or the contrastive estimation for the MI terms. Though learning with different estimation methods, the MI curves for the two estimation methods are close to each other. **Left**: Sprites. **Right**: TIMIT.

Note that S3VAE and R-WAE are both capable of generating recognizable videos, which is validated by our classifier $\mathcal{C}$ with good accuracies. Nevertheless, the results here show that C-DSVAE can learn

Table 5: Ablation study of the augmentations on SMMNIST.

| Methods | Acc↑ | IS↑ | H(y|x)↓ | H(y)↑ |
|---|---|---|---|---|
| DSVAE | 88.19% | 6.210 | 0.185 | 2.011 |
| C-DSVAE w/o content aug | 92.02% | 6.360 | 0.184 | 2.083 |
| C-DSVAE w/o motion aug | 95.25% | 6.471 | 0.175 | 2.093 |
| C-DSVAE | 97.84% | 7.163 | 0.145 | 2.176 |

Table 6: Ablation study of the augmentations on TIMIT.

| Methods | content EER↓ | motion EER↑ |
|---|---|---|
| DSVAE | 5.64% | 19.20% |
| C-DSVAE w/o content aug | 5.09% | 24.30% |
| C-DSVAE w/o motion aug | 4.31% | 31.09% |
| C-DSVAE | 4.03% | 31.81% |

better disentangled factors for the high-quality generation. Results of additional generation tasks can be found in Appendix D, where we swap the latent factors from different sequences.

### 4.6 Other analysis

As mentioned in Sec 2, we attempt to estimate the MI terms using either MWS or the contrastive estimation with augmentations. In Figure 6, we show the MWS estimation of $I(s; x_{1:T})$ improves when the C-DSVAE training objective optimizes the contrastive estimation of $I(s; x_{1:T})$. This means the contrastive estimation could be a surrogate for MI in generative models, and its additional inductive bias can be a main contributor to the superior performance (see Appendix F for quantitative comparisons).

Regarding the augmentations, it is intuitive to adopt both the content augmentation and the motion augmentation. Having both of them can not only boost the learning of the dynamic or static factors, but also improves the disentanglement. Adding only one of them could lead to unbalanced or entangled representation learning. As shown in Table 5, content augmentation alone can improve the accuracy by 3.8%, and motion augmentation alone can improve the accuracy by 7.1%. Two augmentations together can bring a gain of 9.1%. Similarly, in Table 6, though adding only content augmentation or motion augmentation can outperform DSVAE, combining them leads to the largest gain.

Furthermore, in Appendix E.1, we show that even if we don't include $I_q(s, z_{1:T})$ in the optimization, training on just the contrastive terms with augmentations can already lead to the decrease of $I_q(s, z_{1:T})$. In Appendix G, the transition of digits illustrates our learnt latent space is smooth and meaningful.

## 5 Conclusion

In this paper, we propose Contrastively Disentangled Sequential Variational Autoencoder (C-DSVAE) to learn disentangled static and dynamic latent factors for sequence data without external supervision. Our learning objective is a novel ELBO derived differently from prior works, and naturally encourages disentanglement. C-DSVAE uses contrastive estimations of the MI terms to further inject the inductive biases. Our method achieves the state-of-the-art performance on multiple datasets, in terms of the disentanglment metrics and the generation quality. In the future, we plan to extend C-DSVAE to other domains, such as text, biology, agriculture and weather prediction. We will also improve our model's ability to capture long-range dependencies.

## 6 Acknowledgment

This work is supported by Hatch Federal Capacity Funds "3110006036 Hatch Bobea NYC-121437".

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
