# A Proofs

## A.1 Derivation of the per-sequence ELBO

For the input sequence $x_{1:T}$, we show the ELBO derived from the approximate posterior is a lower estimate of its log-likelihood. This proof below is adapted from the standard VAE framework [28, 33], noticing that either the prior or the approximate posterior factorizes over $s$ and $z_{1:T}$.

$$
\begin{aligned}
&\log p(x_{1:T}) \\
\geq & -KL[q(s, z_{1:T}|x_{1:T})||p(s, z_{1:T}|x_{1:T})] + \log p(x_{1:T}) \\
= & \mathbb{E}_{q(s,z_{1:T}|x_{1:T})}\left[\log p(s, z_{1:T}|x_{1:T}) - \log q(s, z_{1:T}|x_{1:T}) + \log p(x_{1:T})\right] \\
= & \mathbb{E}_{q(s,z_{1:T}|x_{1:T})}\left[\log p(x_{1:T}|s, z_{1:T}) - \log q(s, z_{1:T}|x_{1:T}) + \log p(s, z_{1:T})\right] \\
= & \mathbb{E}_{q(s,z_{1:T}|x_{1:T})}\left[\log p(x_{1:T}|s, z_{1:T}) - \log q(s|x_{1:T}) - \log p(z_{1:T}|x_{1:T}) + \log p(s) + \log p(z_{1:T})\right] \\
= & \mathbb{E}_{q(z_{1:T},s|x_{1:T})}\left[\log p(x_{1:T}|s, z_{1:T})\right] - KL[q(s|x_{1:T})||p(s)] - KL[q(z_{1:T}|x_{1:T})||p(z_{1:T})].
\end{aligned}
$$

## A.2 Proof of Theorem 1

Let $p_D$ be the empirical data distribution, assigning probability mass $1/N$ for each of the $N$ training sequences $D$. Define the aggregated posteriors as follows:

$$
q(s) = \mathbb{E}_{x_{1:T}\sim p_D}[q(s|x_{1:T})] = \frac{1}{N}\sum_{x_{1:T}\in D} q(s|x_{1:T}),
$$

$$
q(z_{1:T}) = \mathbb{E}_{x_{1:T}\sim p_D}[q(z_{1:T}|x_{1:T})] = \frac{1}{N}\sum_{x_{1:T}\in D} q(z_{1:T}|x_{1:T}),
$$

$$
q(s, z_{1:T}) = \mathbb{E}_{x_{1:T}\sim p_D}[q(s|x_{1:T})q(z_{1:T}|x_{1:T})] = \frac{1}{N}\sum_{x_{1:T}\in D} q(s|x_{1:T})q(z_{1:T}|x_{1:T}).
$$

With these definitions, we have

$$
\begin{aligned}
&\mathbb{E}_{x_{1:T}\sim p_D}[KL[q(s|x_{1:T})||p(s)]] \\
= & \mathbb{E}_{x_{1:T}\sim p_D}\mathbb{E}_{q(s|x_{1:T})}[\log q(s|x_{1:T}) - \log q(s) + \log q(s) - \log p(s)] \\
= & \mathbb{E}_{q(x_{1:T},s)}\log\left[\frac{q(s|x_{1:T})}{q(s)}\right] + \mathbb{E}_{q(x_{1:T},s)}[\log q(s) - \log p(s)] \\
= & I_q(x_{1:T}; s) + KL[q(s)||p(s)].
\end{aligned}
\tag{9}
$$

In other words,

$$
KL[q(s)||p(s)] = \mathbb{E}_{x_{1:T}\sim p_D}[KL[q(s|x_{1:T})||p(s)]] - I_q(x_{1:T}; s).
\tag{10}
$$

Similarly, we have

$$
KL[q(z_{1:T})||p(z_{1:T})] = E_{x_{1:T}\sim p_D}[KL[q(z_{1:T}|x_{1:T})||p(z_{1:T})]] - I_q(x_{1:T}; z_{1:T}).
\tag{11}
$$

We are now ready to prove the theorem. We derive a dataset ELBO by subtracting a different KL-divergence from the data log-likelihood:

$$
\frac{1}{N} \sum_{x_{1:T} \in D} \log p(x_{1:T}) = \mathbb{E}_{x_{1:T} \sim p_D}[\log p(x_{1:T})]
$$

$$
\geq \mathbb{E}_{x_{1:T} \sim p_D}[\log p(x_{1:T}) - KL[q(s, z_{1:T})||p(s, z_{1:T}|x_{1:T})]]
$$

$$
= \mathbb{E}_{x_{1:T} \sim p_D}[\mathbb{E}_{q(s,z_{1:T}|x_{1:T})}[\log p(x_{1:T}) - (\log q(s, z_{1:T}) - \log p(s, z_{1:T}|x_{1:T}))]]
$$

$$
= \mathbb{E}_{x_{1:T} \sim p_D}[\mathbb{E}_{q(s,z_{1:T}|x_{1:T})}[\log p(x_{1:T}) - \log q(s, z_{1:T}) + \log p(s, z_{1:T}|x_{1:T})]]
$$

$$
= \mathbb{E}_{x_{1:T} \sim p_D}[\mathbb{E}_{q(s,z_{1:T}|x_{1:T})}[\log p(x_{1:T}) - \log q(s, z_{1:T})
$$
$$
+ \log p(x_{1:T}|s, z_{1:T}) + \log p(s, z_{1:T}) - \log p(x_{1:T})]]
$$

$$
= \mathbb{E}_{x_{1:T} \sim p_D}[\mathbb{E}_{q(s,z_{1:T}|x_{1:T})}[\log p(x_{1:T}|s, z_{1:T}) - \log q(s, z_{1:T}) + \log p(s, z_{1:T})]]
$$

$$
= \mathbb{E}_{x_{1:T} \sim p_D}[\mathbb{E}_{q(s,z_{1:T}|x_{1:T})}[\log p(x_{1:T}|s, z_{1:T})]] - \mathbf{KL}[\mathbf{q}(\mathbf{s}, \mathbf{z_{1:T}})||\mathbf{p}(\mathbf{s}, \mathbf{z_{1:T}})]
$$

$$
= \mathbb{E}_{x_{1:T} \sim p_D}[\mathbb{E}_{q(s,z_{1:T}|x_{1:T})}[\log p(x_{1:T}|s, z_{1:T})]]
$$
$$
- \mathbb{E}_{x_{1:T} \sim p_D}[\mathbb{E}_{q(s,z_{1:T}|x_{1:T})}[\log q(s, z_{1:T}) - \log p(s, z_{1:T})]]
$$

$$
= \mathbb{E}_{x_{1:T} \sim p_D}[\mathbb{E}_{q(s,z_{1:T}|x_{1:T})}[\log p(x_{1:T}|s, z_{1:T})]]
$$
$$
- \mathbb{E}_{x_{1:T} \sim p_D}[\mathbb{E}_{q(s,z_{1:T}|x_{1:T})}[\log q(s, z_{1:T}) - \log q(s)q(z_{1:T}) + \log q(s)q(z_{1:T}) - \log p(s, z_{1:T})]]
$$

$$
= \mathbb{E}_{x_{1:T} \sim p_D}[\mathbb{E}_{q(s,z_{1:T}|x_{1:T})}[\log p(x_{1:T}|s, z_{1:T})]]
$$
$$
- \mathbb{E}_{x_{1:T} \sim p_D}\left[\mathbb{E}_{q(s,z_{1:T}|x_{1:T})}\left[\log \frac{q(s, z_{1:T})}{q(s)q(z_{1:T})} + \log \frac{q(s)q(z_{1:T})}{p(s, z_{1:T})}\right]\right]
$$

$$
= \mathbb{E}_{x_{1:T} \sim p_D}[\mathbb{E}_{q(s,z_{1:T}|x_{1:T})}[\log p(x_{1:T}|s, z_{1:T})]]
$$
$$
- I_q(s; z_{1:T}) - \mathbb{E}_{x_{1:T} \sim p_D}\left[\mathbb{E}_{q(s,z_{1:T}|x_{1:T})}\left[\log \frac{q(s)q(z_{1:T})}{p(s)p(z_{1:T})}\right]\right]
$$

$$
= \mathbb{E}_{x_{1:T} \sim p_D}[\mathbb{E}_{q(s,z_{1:T}|x_{1:T})}[\log p(x_{1:T}|s, z_{1:T})]] - I_q(s; z_{1:T})
$$
$$
- \mathbb{E}_{x_{1:T} \sim p_D}\left[\mathbb{E}_{q(s,z_{1:T}|x_{1:T})}\left[\log \frac{q(s)}{p(s)}\right]\right] - \mathbb{E}_{x_{1:T} \sim p_D}\left[\mathbb{E}_{q(s,z_{1:T}|x_{1:T})}\left[\log \frac{q(z_{1:T})}{p(z_{1:T})}\right]\right]
$$

$$
= \mathbb{E}_{x_{1:T} \sim p_D}[\mathbb{E}_{q(s,z_{1:T}|x_{1:T})}[\log p(x_{1:T}|s, z_{1:T})]] - I_q(s; z_{1:T}) - KL[q(s)||p(s)] - KL[q(z_{1:T})||p(z_{1:T})]
$$

$$
= \mathbb{E}_{x_{1:T} \sim p_D}[\mathbb{E}_{q(s,z_{1:T}|x_{1:T})}[\log p(x_{1:T}|s, z_{1:T})]] - I_q(s; z_{1:T})
$$
$$
- (\mathbb{E}_{x_{1:T} \sim p_D}[KL[q(s|x_{1:T})||p(s)]] - I_q(s; x_{1:T}))
$$
$$
- (\mathbb{E}_{x_{1:T} \sim p_D}[KL[q(z_{1:T}|x_{1:T})||p(z_{1:T})]] - I_q(z_{1:T}; x_{1:T}))
$$

$$
= \mathbb{E}_{x_{1:T} \sim p_D}[\mathbb{E}_{q(z_{1:T}, s|x_{1:T})}[\log p(x_{1:T}|s, z_{1:T})]
$$
$$
- \mathbb{E}_{x_{1:T} \sim p_D}[KL[q(s|x_{1:T})||p(s)]] - \mathbb{E}_{x_{1:T} \sim p_D}[KL[q(z_{1:T}|x_{1:T})||p(z_{1:T})]]
$$
$$
+ I_q(s; x_{1:T}) + I_q(z_{1:T}; x_{1:T}) - I_q(s; z_{1:T}).
$$

$$(12)$$

where we have plugged in (10) and (11) in the third to last step. The last equation of (12) is the dataset ELBO objective in the main text.

## A.3 MWS estimation of $I(s; z_{1:T})$

We estimate $I(s; z_{1:T})$ using minibatch weighted sampling (MWS). The estimation is simply adapted from [23, 33]:

$$
I(s; z_{1:T}) = H(s) + H(z_{1:T}) - H(s, z_{1:T})
$$

$$
H(s) = -\mathbb{E}_{q(s, z_{1:T})}[\log q(s)] \approx -\frac{1}{M} \sum_{i=1}^{M}\left[\log \sum_{j=1}^{M} q(s(x_{1:T}^i)|x_{1:T}^j) - \log(NM)\right]
$$

$$(13)$$

where $N$ is the dataset size and $M$ is the minibatch size. $H(z_{1:T}), H(s, z_{1:T})$ are estimated similarly, with the trajectory of $z_{1:T}$ sampled for each sequence in the minibatch.

## A.4 Contrastive estimation of MI

Suppose $z_{1:T}$ is derived from the anchor sequence $x_{1:T}$, and we have one "positive" sequence $x_{1:T}^+$ as well as $n$ "negative" sequences $\{x_{1:T}^j\}_{j=1}^n$ in total. We have

$$
\begin{aligned}
&\mathbb{E}_{p_D}\left[\log \frac{\frac{q(x_{1:T}^+|z_{1:T})}{q(x_{1:T}^+)}}{\frac{q(x_{1:T}^+|z_{1:T})}{q(x_{1:T}^+)} + \sum_{j=1}^n \frac{q(x_{1:T}^j|z_{1:T})}{q(x_{1:T}^j)}}\right]\\
&= -\mathbb{E}_{p_D}\left[\log(1 + \frac{q(x_{1:T}^+)}{q(x_{1:T}^+|z_{1:T})}\sum_{j=1}^n \frac{q(x_{1:T}^j|z_{1:T})}{q(x_{1:T}^j)})\right]\\
&\approx -\mathbb{E}_{p_D}\left[\log\left(1 + \frac{q(x_{1:T}^+)}{q(x_{1:T}^+|z_{1:T})}\cdot n\mathbb{E}_{x_{1:T}^j}\left[\frac{q(x_{1:T}^j|z_{1:T})}{q(x_{1:T}^j)}\right]\right)\right]\\
&= -\mathbb{E}_{p_D}\left[\log\left(1 + \frac{q(x_{1:T}^+)}{q(x_{1:T}^+|z_{1:T})}\cdot n\right)\right]\\
&= -\mathbb{E}_{p_D}\left[\log\left(\frac{1}{1+n} + \frac{q(x_{1:T}^+)}{q(x_{1:T}^+|z_{1:T})}\cdot\frac{n}{1+n}\right)\right] - \log(n+1)\\
&\leq -\mathbb{E}_{p_D}\left[\frac{1}{n+1}\log 1 + \frac{n}{n+1}\log\frac{q(x_{1:T}^+)}{q(x_{1:T}^+|z_{1:T})}\right] - \log(n+1)\\
&= -\frac{n}{n+1}E_{p_D}\left[\log\frac{q(x_{1:T}^+)}{q(x_{1:T}^+|z_{1:T})}\right] - \log(n+1)\\
&= \frac{n}{n+1}\mathbb{E}_{p_D}\left[\log\frac{q(x_{1:T}^+|z_{1:T})}{q(x_{1:T}^+)}\right] - \log(n+1)\\
&\approx \frac{n}{n+1}I(x_{1:T}; z_{1:T}) - \log(n+1)
\end{aligned}
\tag{14}
$$

where the first $\leq$ step uses Jensen's inequality, and the approximations by sampling follow the development of CPC [2]. Similar derivations can also be obtained for $s$.

**Use augmentation in contrastive estimation** Note that in CPC, for each frame, the positive example is a nearby frame. That is, CPC uses the temporal smoothness as the inductive bias for learning per-frame representations. In our setup however, we treat the entire trajectory $z_{1:T}$ as samples in contrastive estimation, and we must resort to other inductive biases for finding positive examples.

Here we provide another motivation for the use of augmented sequences as positive examples. Imagine that the latent factors are discrete, and the mapping from the latent factors $(s, z_{1:T})$ to observations $x_{1:T}$ is done by a deterministic mapping (while the factors remain random variables with prior distributions). Furthermore, the mapping is invertible in the sense that we could identify a unique $(s, z_{1:T})$ that generates $x_{1:T}$. Then for two sequences $x_{1:T}^1$ and $x_{1:T}^2$ generated with common dynamic factors $z_{1:T}^*$ but different static factors $s^1$ and $s^2$, we have

$$
\begin{aligned}
p(x_{1:T}^1) &= p(z_{1:T} = z_{1:T}^*, s = s^1) = p(z_{1:T} = z_{1:T}^*)\,p(s = s^1),\\
p(x_{1:T}^1|z_{1:T}) &= 1(z_{1:T} = z_{1:T}^*)\,p(s = s^1),\\
\frac{p(x_{1:T}^1|z_{1:T})}{p(x_{1:T})} &= \frac{1(z_{1:T} = z_{1:T}^*)}{p(z_{1:T} = z_{1:T}^*)} = \frac{p(x_{1:T}^2|z_{1:T})}{p(x_{1:T})}.
\end{aligned}
$$

The last equation is obtained by dividing the second line by the first line, where the $p(s = s^1)$ is conveniently canceled out. In other words, under the above simplifying assumptions, the probability ratio used in contrastive estimation is the same for the original sequence and the augmented sequence.

## A.5 Summary of objective function and estimations

Though the objective function (12) has been proven to be an ELBO, we don't have to stick to the natural coefficients. For example, to control the information bottleneck, one can add coefficients to

the KL terms. To improve the disentanglement, one can add coefficients to the MI terms. This leads to the final training objective (6).

In practice, $\gamma = 1$ gives good results and we mainly tune $\alpha, \beta$. As mentioned in Sec 2.3, $I(s; x_{1:T}), I(z_{1:T}; x_{1:T})$ are estimated contrastively. $I(s; z_{1:T})$ is estimated through mini-batch weighted sampling (MWS) as discussed in A.3. Note that we use contrastive learning here to maintain some invariance (static or dynamic factors) across different views, besides just estimating the MI terms. For tasks where no such invariance exists, MWS is still a good option for the estimation (e.g. $I(z_{1:T}; s)$). More comparisons can be found in F.

## B  Disentanglement Metrics

**Accuracy** (Acc) measures how well the fixed factor can be identified by the classifier $C$. Given an input sequence $x_{1:T}$, the encoder would produce $z_{1:T}$ and $s$. If we randomly sample from the prior instead of the posterior of $s$ for decoding/generation, the classifier $C$ should still recognize the same motion captured by $z_{1:T}$ while identifying different contents induced by the random $s$. For example, in the Sprites dataset, if we randomly sample $s$ from the prior and fix $z_{1:T}$, we should see the generated characters with different colors performing the same action in the same direction.

**Inception Score** ($IS$) computes the KL-divergence between the conditional predicted label distribution $p(y|x_{1:T})$ and marginal predicted label distribution $p(y)$ from $C$, $IS = \exp(\mathbb{E}_{p(x)}[KL[p(y|x)||p(y)]])$. $y$ is the predicted attribute such as color, action, expression, etc. $p(y|x_{1:T})$ is usually taken from the logits after the softmax layer. $p(y)$ is the marginalization across all the test samples, $p(y) = \frac{1}{N}\sum_{i=1}^{N} p(y|x)$. Since we hope the generated samples to be as diverse as possible, $IS$ is expected to be high.

**Inter-Entropy** is similar to $IS$ but measures the diversity solely on the marginal predicted label distribution $p(y)$, $H(y) = -\sum_y p(y)\log p(y)$. The higher $H(y)$ is, the more diverse generated sequences would be.

**Intra-Entropy** measures the entropy over $p(y|x)$, $H(y|x) = -\sum_y p(y|x)\log p(y|x)$. Lower intra-entropy means more confident predictions.

**Equal Error Rate** (EER) is only used in the TIMIT experiments for audio evaluation. It means the common value when the false rejection rate is equal to the false acceptance rate.

## C  Training Details

### C.1  Dataset

**Sprites** is a cartoon character video dataset. Each character's (dynamic) motion is captured by three actions (walking, spellcasting, slashing) and three directions (left, front, right). The (static) content comprises of each character's skin color, tops color, pants color and hair color. Each color has six variants. Each sequence is composed of 8 frames of RGB images with size $64 \times 64$. There are in total 1296 characters. 1000 of them are used for training and validation. The rest 296 characters are used for testing.

**MUG** is a facial expression video dataset. The static factor corresponds to the person's identity. Each individual performs six expressions (motion): anger, fear, disgust, happiness, sadness and surprise. Each sequence contains 15 frames of RGB images with size $64 \times 64$ (after resizing). There are in total 3528 videos with 52 people. Each video contains 50 to 160 frames. Length-15 clips are sampled from the raw videos. Each frame from the raw videos is reshaped to $64 \times 64$.

**SM-MNIST** (Stochastic Moving MNIST) records the random movements of two digits. Each sequence contains 15 gray scale images of size $64 \times 64$. Individual digits are collected from MNIST. We adopt the S3VAE setup with two moving digits instead of just one.

**TIMIT** is a corpus of read speech for acoustic-phonetic studies and speech recognition. The utterances are produced by American speakers of eight major dialects reading phonetically rich sentences. It contains 6300 utternaces (5.4 hours) with 10 sentences from each of 630 speakers. The data preprocessing follows prior works. We extract spectrogram features (with a frame shift size

of 10ms) from the audio, and the segments of 200ms duration (20 frames) are sampled from the utterances which are treated as independent sequences for learning.

## C.2 Model architecture

Each frame is passed to an encoder first to extract an abstract feature as the input for LSTM. Such encoder is a convolutional neural network with 5 layers of channels [64, 128, 256, 512, 128]. Every layer uses a kernel with size 4 followed by BatchNorm and LeakyReLU (except the last layer using Tanh). The dimensionality of each abstract feature is 128. The decoder is also a convolutional neural network, with 5 hidden layers of channels [512, 256, 128, 128, 64]. Each layer is followed by BatchNorm and ReLU. The whole architecture is consistent with S3VAE. For the audio dataset, we use the same architecture as DSVAE. A bi-LSTM with hidden size 256 takes the 128d features as inputs to produce $\mu$ and $\sigma$ for $s$ with size $d_s$ based on the last cell state. Another uni-LSTM produces $\mu$ and $\sigma$ for $z_{1:T}$ at each time step with dimension $d_m$ each. We set $d_s = 256, d_m = 32$ in Sprites, $d_s = 128, d_m = 8$ in MUG, $d_s = 256, d_m = 32$ in SM-MNIST, $d_s = 256, d_m = 64$ in TIMIT. $\alpha = 0.9$ or $1.0$ can produce good results for all datasets. $\beta = 1.0, 0.5, 5.0, 1.0$ are set respectively for Sprites, MUG, SM-MNIST, TIMIT. The dynamic prior is parameterized by an LSTM with hidden size 256.

The learning rate is set to be 0.001 for MUG, SM-MNIST and TIMIT. Sprites takes a learning rate 0.0015. C-DSVAE is trained with up to 1000 epochs (typically much fewer) on one NVIDIA V100 GPU.

## D Supplementary Experiments

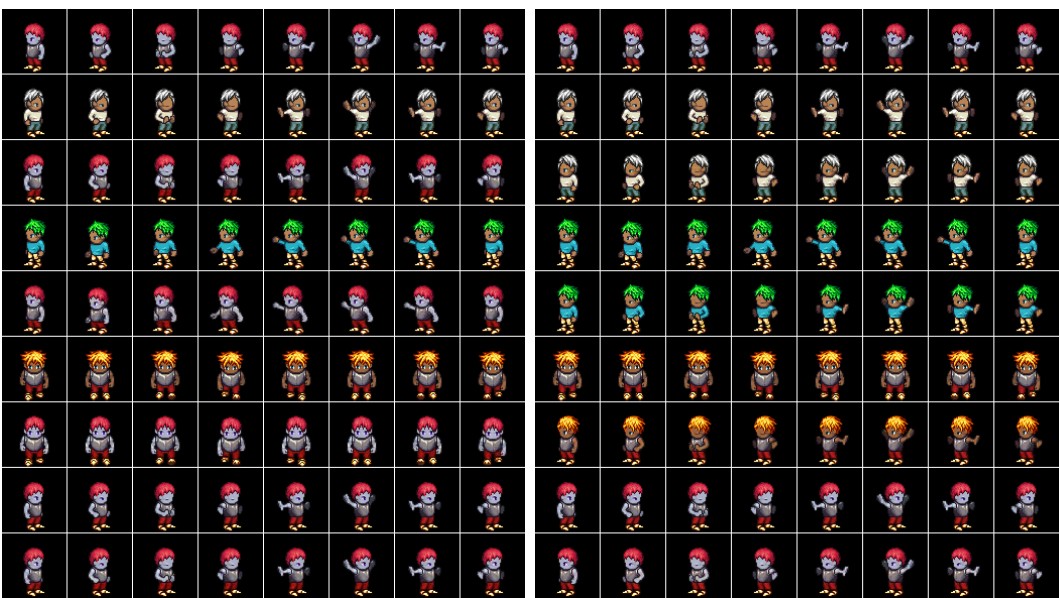

Figure 7: Row 1, 2, 4, 6, 8 are the raw test input sequences. **Left:** For row $i$ ($i$ is odd and $i > 1$), $s$ is set to be the same as the $s$ from row 1, while $z_{1:T}$ of each row is retained. **Right:** For row $i$ ($i$ is odd and $i > 1$), $z_{1:T}$ is set to be the same as the $z_{1:T}$ from row 1, while $s$ of each row is retained.

## D.1 Sprites

We show more experiments of fixing one factor and replacing the other on Sprites in Figure 7. In addition, Figure 8 shows the generated sequences when the factors are sampled from the priors (either static or dynamic factors).

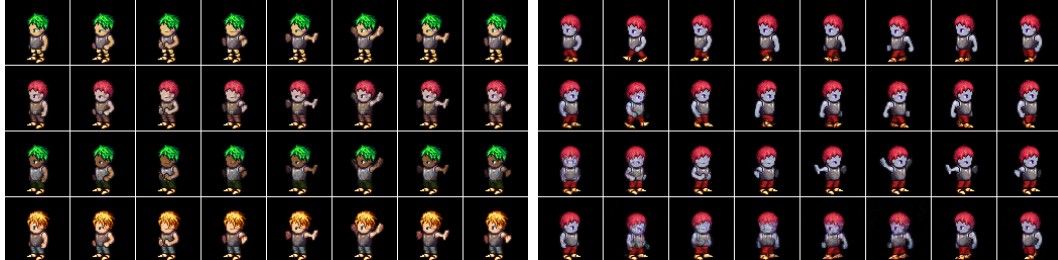

Figure 8: The random generations of contents and motions. **Left:** $z_{1:T}$ from row 1 is fixed and $s$ is sampled from $p(s)$ for other rows. **Right:** $s$ from row 1 is fixed and $z_{1:T}$ is sampled from $p(z_{1:T})$ for other rows.

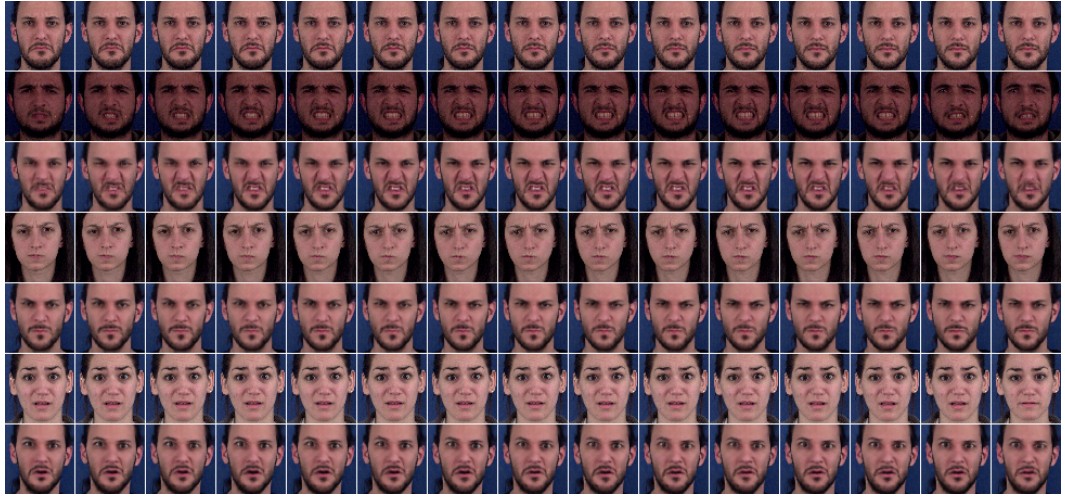

(a) Fix the content. Row 1, 3, 5, 7 use the same $s$. Meanwhile, they also take $z_{1:T}$ from the previous row for generation. The person identity is well-preserved with different expressions.

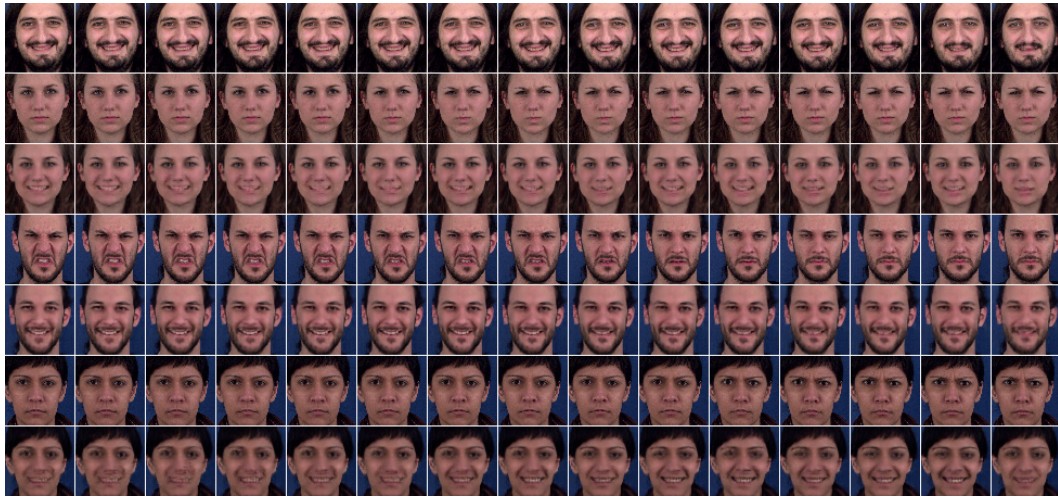

(b) Fix the motion. Row 1, 3, 5, 7 use the same $z_{1:T}$. Meanwhile, they also take $s$ from the previous row for generation. All the faces have the same expression and the identity is consistent with the previous row.

Figure 9: Fix either the static or dynamic factors and replace the other.

## D.2 MUG

Figure 9 shows the generated sequences when we fix either the motion or content factors. In Figure 9a, the person identity is consistent with row 1 but has the same expression as the previous row. In Figure 9b, the person identity is consistent with the previous row but all with the same expression as row 1.

## D.3 SM-MNIST

Figure 10: Generate different contents. For every two rows separated by the red lines, the first row is from the test set and the second row inherits the dynamic factors $z_{1:T}$ from the first row, while samples $s$ from prior $p(s)$. As a result, the same motion is preserved with different digits.

Figure 10 generates random digits from $p(s)$ to replace the content from the raw test input sequences. The motions are well-preserved while the contents are totally different.

On the other hand, Figure 11 preserves the content digits and randomly sample $p(z_{1:T})$. The motions of the digits then become different.

Another interesting generation task is swapping, as shown in Figure 12. Given two input sequences, we exchange their dynamic and static factors.

## D.4 TIMIT

Figure 13 shows a cross-reconstruction of different audio clips. Each heatmap if of dimension $80 \times 20$ which corresponds to an audio clip of length 200ms. The 80d feature at each time step is the mel-scale filter bank feature. The static factor from the first row and the dynamic factors from the first column are mixed for the generation. As we can observe, the linguistic contents are kept the same along each column and the timbres reflected as the harmonics are mostly preserved along each row.

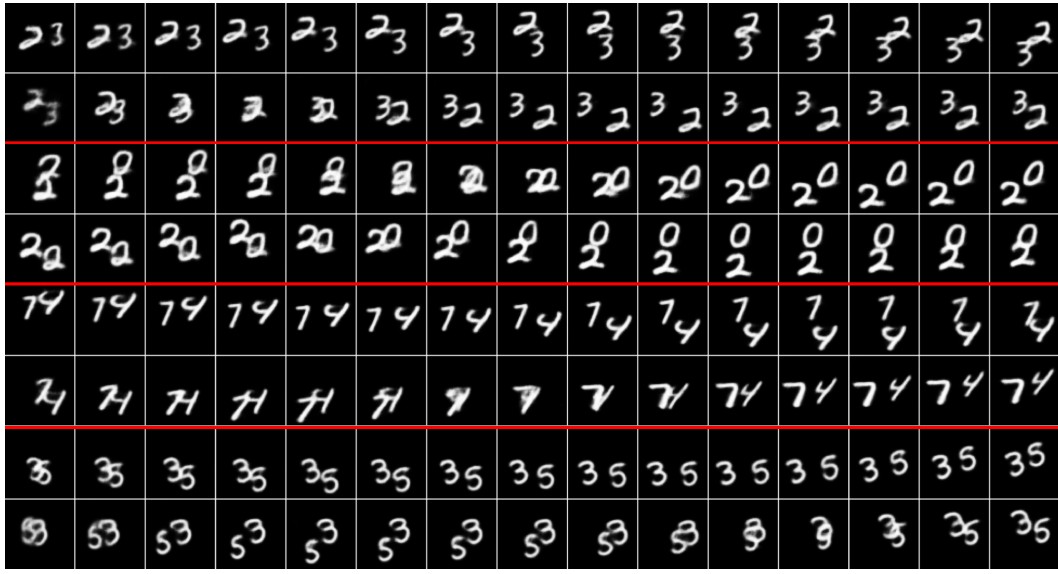

Figure 11: Generate different motions. For every two rows separated by the red lines, the first row is from the test set and the second row inherits the static factor $s$. $z_{1:T}$ is sampled from the prior $p(z_{1:T})$. As a result, the same digits are preserved with different movements.

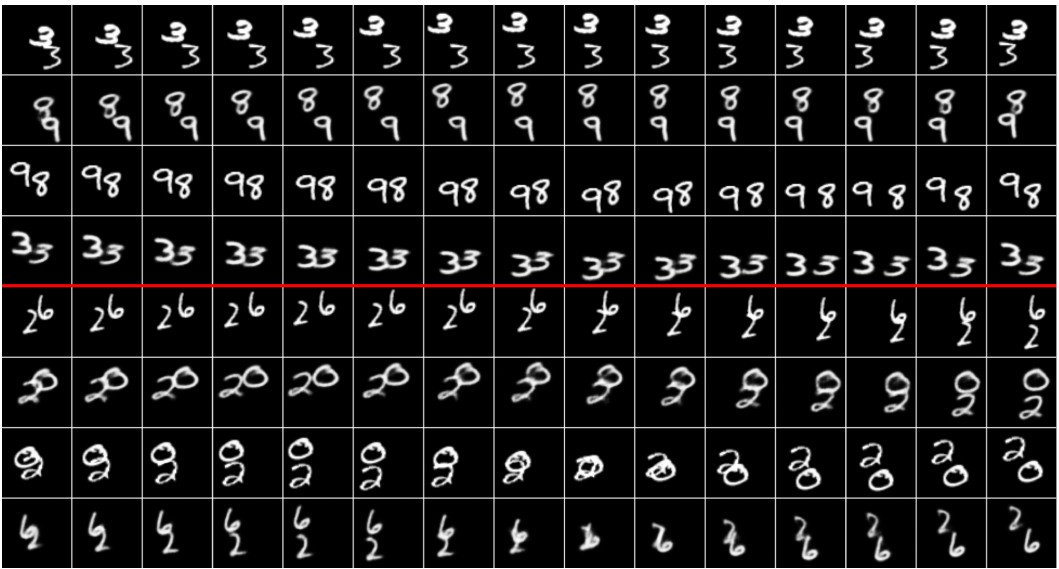

Figure 12: Swap the static and dynamic factors. For every 4 rows separated by the red line, row 1 and row 3 are the raw test sequences with different contents and motions. Row 2 takes $s$ from row 3 and $z_{1:T}$ from row 1. Row 4 takes $s$ from row 1 and $z_{1:T}$ from row 3. We present 2 such swapping sets: row 1~4, row 5~8.

# E    Observations on mutual information (MI)

## E.1    MI between $s$ and $z_{1:T}$

Figure 14a demonstrates the curve of MI $I(s; z_{1:T})$ during training on Sprites when we don't include $I(s; z_{1:T})$ in the objective function. We estimate $I(s; z_{1:T})$ using the Minibatch Weighted Sampling (MWS) estimation (though not optimized directly). The curve increases in the early stage of training, implying that while learning useful representations for reconstruction, disentanglement is compromised in the early stage. But as the training continues, the model picks up the disentanglement

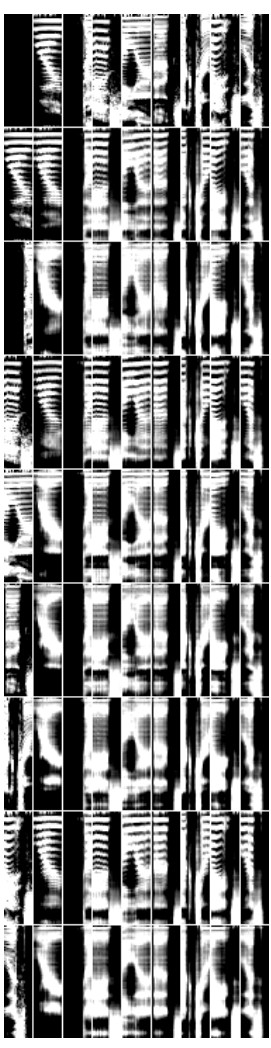

Figure 13: The cross generation of 8 audio clips. Plot at $(i+1)$-th row and $(j+1)$-th column reconstructs the $i$-th static factor and $j$-th dynamic factors.

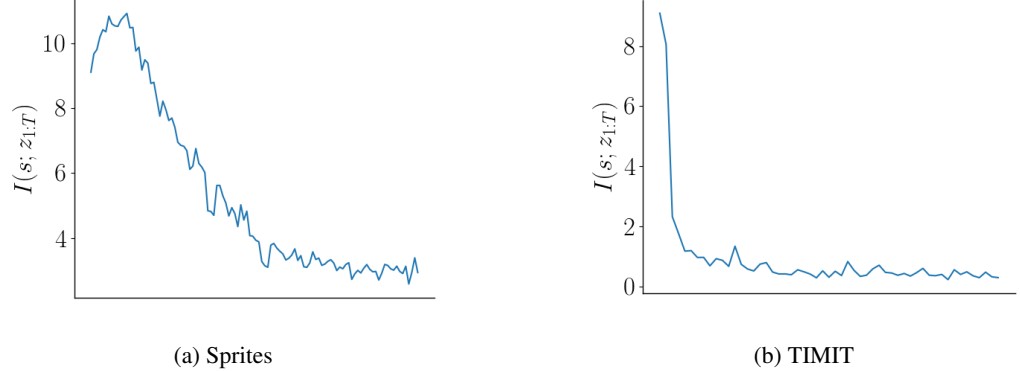

(a) Sprites

(b) TIMIT

Figure 14: The mutual information $I(s; z_{1:T})$ estimated by MWS decreases during the training even if we don't include $I(s; z_{1:T})$ in the objective.

terms besides the reconstruction. Figure 14b demonstrates the same experiment on another dataset TIMIT.

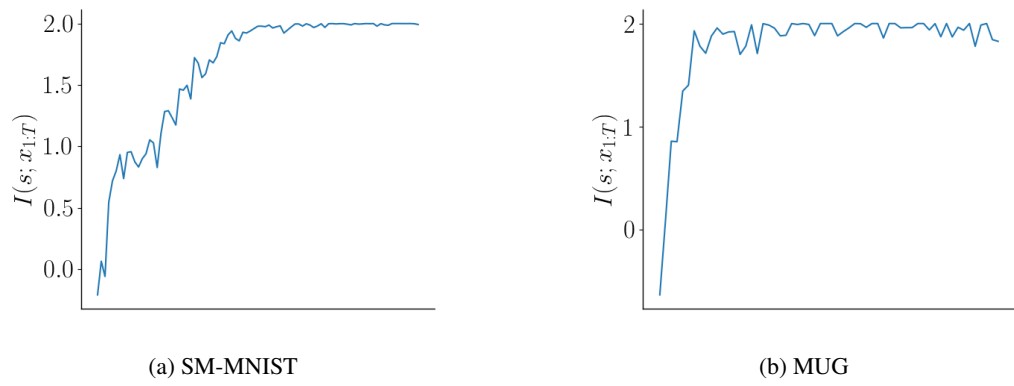

(a) SM-MNIST          (b) MUG

Figure 15: The mutual information $I(s, x_{1:T})$ estimated by MWS increases during the training, demonstrating the effectiveness of the contrastive estimation.

## E.2    MI between $s$ and $x_{1:T}$

We show in Figure 15 that, the contrastive estimation could boost $I(s; x_{1:T})$ during training. Our training process optimizes the contrastive estimation, but as we can see in the figure, the MI term $I(s; x_{1:T})$ estimated by MWS also increases accordingly.

## F    Ablation Study on Estimation Methods

Figure 16: Interpolation in the latent space. For every 5 rows separated by the red line, the first and last rows have different contents $s$, but share the same motion. The 3 rows in between keep the same motion but their $s$'s linearly interpolate between row 1 and row 5. One can observe that the content transition is smooth, while the motion is intact.

Our C-DSVAE estimates $I(s; x_{1:T})$, $I(z_{1:T}; x_{1:T})$ with contrastive learning and $I(s; z_{1:T})$ with MWS. To further demonstrate the advantage of the contrastive estimation, we compare it with the

Table 7: Compare MWS and contrastive estimation of $I(s; x_{1:T})$ and $I(z_{1:T}; x_{1:T})$ on MUG. "all MWS est" means that C-DSVAE optimizes MI terms all estimated by MWS rather than the contrastive estimation.

| Methods | Acc↑ | IS↑ | H(y\|x)↓ | H(y)↑ |
|---|---|---|---|---|
| DSVAE | 54.29% | 3.608 | 0.374 | 1.657 |
| DSVAE+all MWS est | 66.25% | 4.796 | 0.175 | 1.743 |
| C-DSVAE | 81.16% | 5.341 | 0.092 | 1.775 |

Table 8: Compare MWS and contrastive estimations of $I(s; x_{1:T})$ and $I(z_{1:T}; x_{1:T})$ on SM-MNIST.

| Methods | Acc↑ | IS↑ | H(y\|x)↓ | H(y)↑ |
|---|---|---|---|---|
| DSVAE | 88.19% | 6.210 | 0.185 | 2.011 |
| DSVAE+all MWS est | 91.81% | 6.312 | 0.205 | 2.107 |
| C-DSVAE | 97.84% | 7.163 | 0.145 | 2.176 |

Table 9: Performance with different batch sizes on TIMIT.

| batch size | content EER↓ | motion EER↑ |
|---|---|---|
| 64 | 4.21% | 30.23% |
| 128 | 4.07% | 31.42% |
| 256 | 4.03% | 31.81% |

model with MWS estimations on all the MI terms including $I(s; x_{1:T})$, $I(z_{1:T}; x_{1:T})$. Table 7 and 8 give the observations. With all MWS estimations, the performance would drop heavily.

These results help support the claim that the inductive biases brought by contrastive learning might contribute more to the good performance than the MI estimation. Also note that contrastive learning works well when the goal of learning is to maintain some invariance between different views. For other MI estimation tasks, contrastive learning might not be the best option.

## G  Interpolation in Latent Space

To further show our learnt latent space is smooth and meaningful, we linearly interpolate between 2 content factors corresponding to different digit pairs and generate the sequences. In Figure 16, one can see that the transition from one content to another is smooth. In the first example, ("5", "4") gradually changes to ("8", "8"). In the second example, ("0", "2") gradually changes to ("9", "1"). While "2" transforms to "1", it first becomes "7" and its font gets slimmer.

## H  Sensitivity Analysis on Batch Size

In Table 9, we show how different batch sizes would affect the evaluation performance on TIMIT. Batch size 256 gives the best numbers.