# OpenReview forum: "Contrastively Disentangled Sequential  Variational Autoencoder"
_NeurIPS.cc/2021/Conference — NeurIPS 2021 Poster_

### Official Review · Reviewer_wLW8 · 2021-07-14

**Rating:** 7
**Confidence:** 4

**Summary:**

The focus of this work is disentangled representation in sequential data. It merges existing approaches/views of contrastive learning, disentanglement, and dynamical data modeling, into a unified framework with the goal to extract better representations of sequential data (video, audio experiments).
The paper is well written and self-contained which makes it easy to follow. The authors made an effort to present related literature (with useful remarks on differences) and in the experiments, they compare to the most relevant state-of-art approaches.
Many insightful results are redelegated to the supplementary even though the paper could have content on 9 pages (submitted version has 8).  Although this paper builds on previous works, the detailed analysis of both theoretical and empirical findings makes their contribution in terms of contrastive disentanglement valuable to the community.

**Limitations And Societal Impact:**

There is no potential negative societal impact of this work.

**Main Review:**

From the ablation study in F.1 seems like motion augmentation is more important than content augmentation. Do you have any intuition on that? What would be an example dataset where the static augmentation will matter more?

Are there any results that show why stochastic prior for the latent space is better than deterministic one?

Given the generative nature of the model, would it be possible to use C-DSVAE for generating counterfactuals for example? I assume this will depend on how much control you have on the generative process, if besides disentangling static from dynamic factors, could you control for some factorized variables.

Since the approach relies on inductive biases, what would be the implication of changing the current ones with for example physics-inspired inductive biases? The proposed contrastive estimation with augmentation is tailored to video and audio data and I wonder how this could be generalized for datasets of different nature. Is there any alternative that can be considered?

Some relevant references for deep learning approaches to state-space models could be mentioned, such as Kalman VAEs [1]  which also split the learning mechanics into static and dynamic.


[1] Fraccaro, M., Kamronn, S., Paquet, U., & Winther, O. (2017). A disentangled recognition and nonlinear dynamics model for unsupervised learning. arXiv preprint arXiv:1710.05741.

------------------------------------------------------------
Update after Author Feedback and Discussion
------------------------------------------------------------

I thank the authors for the clarification. My questions and concerns have been mostly addressed, and I still argue for acceptance of the paper.  A minor note to avoid ambiguity would be to maybe if not in the title, at least in the abstract the authors should mention that the method (implementation) in the current form, only applies to video and audio data.

**Time Spent Reviewing:**

8

---

> ### Author Response · Authors · 2021-08-10
> **Importance of augmentation and discussion on possible extensions**
>
> Thank you for recognizing our work! We can further clarify some of your concerns.
>
> 1. Perhaps you mean content augmentation is more important? Results without content augmentation are worse than the results without motion augmentation. This is actually expected since reversing the sequence is a clean and effective way to augment the sequence data, while preserving the static content. The motion augmentation also helps in general, as you can see from tables 5 and 6. For instance, in table 5, motion augmentation significantly improves the acc from 88.19% to 92.02%. We think both augmentation are important, although “motion” is more complex in nature and finding a good motion augmentation is generally harder.
> 2. Using stochastic prior naturally allows us to model uncertainty in the dynamics, which is a principle advocated by VAE. And in general variational methods are shown to learn better representations than the deterministic counterparts. In our method, at each time step, the prior depends on the sample of the previous time step (see section 2.1 for the formal definition). Similar stochastic prior has been adopted by FHVAE, DSVAE, S3VAE, RWAE etc, for modeling sequence data. You may also find the motivations/support for using stochastic prior in these references.
> 3. We think our general intuitions can be extended to other scenarios including different application domains (e.g., counterfactuals or physics systems) and different priors and inductive bias. The key to the success of possible extensions would be that we can provide data augmentations that modify one set of latent variables while keeping the others roughly intact (as you hypothesized, it depends on how well one can control the generation).
> 4. Thanks for the pointer. We will include the reference in our revision.

---

### Official Review · Reviewer_RjYQ · 2021-07-15

**Rating:** 6
**Confidence:** 3

**Summary:**

This paper proposes to use contrastive learning methods to improve disentangled sequential representation learning. The proposed C-DSVAE maximizes a novel evidence lower bound that introduces and maximizes the mutual information between latents and input data, both terms are naturally estimated from contrastive loss. The experiments are comprehensive and convincing.

**Limitations And Societal Impact:**

The authors discussed limitations and societal impacts.

**Main Review:**

Strenghs:
1. The paper is well-written and well-organized.
2. The formulation in Equation (5) is very interesting. I like the idea of using contrastive learning to naturally estimate the mututal information between inputs and latents. The same idea can be found in WAEs, but contrastive learning smartly leaverages data augmentation and makes the training easier and more robust.
3. Experiments are extensive and convincing.


Questions:
1. Line 97-98, why this makes the problem "less severe"? Is there any evidence to support that "it is hard for $s$ to capture the dynamics"? How to exclude the possibility of posterior collapse where all information is encoded in $s$ (this is likely if $p(z_{1:T})$ is hard to train, especially like Sprites has only 8 frames)?
2. Evaluation:
    1. Line 261, are '4 5' and '5 4' considered as two combinations? The numbers in Table 3 are quite high, but it seems impossible for a classifier to tell the order of two digits. Can the authors provide more explanation?
    2. Line 294, why reporting MWS if "it may be hard to estimate the distributions accurately" (as stated in line 119)? Can the author provide more typical evaluation metrics such as MIG?
3. Samples:
    1. Can authors provide generation samples from prior on MUG and TIMIT?
    2. Figure 13 is hard to judge, can the authors provide audio samples like gender swapping in [29] (DSVAE)?
4. The results on MUG looks quite sharp and nice. I am curious how the model perform in higher resolution or more complicated video sequence (a slightly harder case would be uncropped MUG faces at higher resolution).
5. Regarding to relating works, I think that the proposed bound is very related to the existing work [\*]. It would be nice to discuss it in the revision of the paper. Specifically, [\*] introduces a similar bound with corresponding MI terms.

[\*] Cheng, Pengyu, et al. "Improving disentangled text representation learning with information-theoretic guidance." arXiv preprint arXiv:2006.00693 (2020).

**Time Spent Reviewing:**

4

---

> ### Author Response · Authors · 2021-08-10
> **Samples are provided and more discussions of motivations are included**
>
> Thanks for recognizing our contributions! The summarization of our work is accurate. Here are some clarifications to your questions:
>
> 1. Regarding the degeneration is “less severe”: it is intuitively true that if the data has high variance in time, modeling it with only a static variable (which has variance 0 in time) will lead to poor likelihood estimation (or per-frame reconstruction). On the other hand, it is very possible that, without additional regularization (e.g., our contrastive MI terms), the learnt motion variables $z$ can carry redundant information of the static variables $s$. And it is the goal of our paper and prior works (DSVAE, S3VAE) to alleviate this issue.
> Regarding “posterior collapsing”: If the reviewer means the learned $z_{1:T}$ can contain little information about input, this degeneracy is avoided in our method by maximizing the $I(z,x)$ term. Also, posterior collapsing tends to happen with extremely powerful decoder networks that can take advantage of large/full context, see, e.g.,
> Bowman et al., 2015. Generating sentences from a continuous space..
> But in our case we use a DNN for per-frame reconstruction to alleviate this issue.
> 2. (1) On SM-MNIST dataset, we followed exactly the same evaluation metric implementation of S3VAE, which views prediction and ground-truth as unordered digit sets (each digit set contains 2 digit). So yes, ‘4 5’ is the same as ‘5 4’. (2) In line 294, we were trying to show that MWS and contrastive estimation are positively correlated, so as estimators of MI itself, they are comparable. (But contrastive estimation has additional inductive bias that leads to better representation quality, see Appendix F.2.)
> Regarding MIG: MIG requires the known ground-truth factors and is harder to estimate. For example, on SM-MNIST, it is hard to define a ground-truth factor for the movements.
> 3. We provide some extra samples in this anonymous github: https://anonymous.4open.science/r/NeurIPS21_rebuttal-4265. “m_to_f” stands for male to female, and “f_to_m” stands for female to male. You can observe that the first harmonics (near bottom) move up in “m_to_f” and move down in “f_to_m”, which is consistent with DSVAE. We also provide some extra samples drawn from the prior for mug and timit. For mug, we extract latent content and motion posteriors from test samples, and replace the content posterior with the content prior to obtain the samples “content_prior_{1,2,3,4,5}”. And we obtain the “motion_prior_{1,2,3,4,5}” similarly by replacing the motion posterior with motion prior. For timit sample, it is best to use mac finder preview to go through the plots.
> 4. High resolution video generation is an interesting future direction for our method. We can explore it in the future.
> 5. Thanks for the pointer and suggested reference. This reference proposes to optimize the same set of mutual information terms similar to our vanilla objective. However, in our paper, we show that they can be naturally derived from the fundamental principle of VAE, and we additionally proposed contrastive estimation and data augmentation to strengthen them, which was not done in the reference. We will discuss the reference in related work in revision.

---

### Official Review · Reviewer_DVU8 · 2021-07-16

**Rating:** 6
**Confidence:** 4

**Summary:**

The paper makes contributions to the research on disentangled sequential representation learning based on Variational Autonecoders (VAES). Towards this end, it

1)  proposes an objective function, which theoretically constitutes a lower bound of the marginal likelihood, while containing interpretable and intuitive mutual information loss terms for encouraging disentanglement between a static and a sequence of dynamic latent factors.

2) proposes a non-parametric estimation of the aforementioned mutual information terms that leverages data-augmentation to further improve performance.

3) demonstrates both quantitative and qualitative performance gains compared to other works on a variety of disentanglement metrics and video datasets.

**Limitations And Societal Impact:**

Yes

**Main Review:**

In general, the paper is easy to follow, and the proposed method is both theoretically and intuitively motivated while tackling an important problem. The results (especially the qualitative of section 4.5) are compelling. However, some points both in the presentation and in the experiments can be improved in order to better demonstrate the proposed method. In particular,

(1) In section 2.3, the implementation of the proposed estimate is not very clear. For example, how many n 'negative sequences are considered and what computational overhead sampling these sequences is it induced compared to the MWS estimate? It would be interesting to demonstrate convergence wrt to n and compare with the MWS estimate. Similarly, for the \phi term how many samples are considered when estimating the aggregated posterior q(x1:T) in the denominator? How the positive sequence x+ is obtained?

(2) More importantly, it is not clear what improvement p  presented in Tables 1-4, is attributed to the augmentation presented in 2.3 and what improvement to the modified ELBO of equation (5). I think the proposed approach would be better demonstrated via an ablation study on at least one dataset that includes :
i) results with the objective in Equation (5) and the MWS estimate,
ii) results with the objective in Equation (5) and the estimate in Equation (6) without data augmentation
iii) performance of one competitive model with data augmentation when applicable

If such an ablation study is provided I would be happy to increase my score.

(3) Regarding the writing,
i. there are several errors that need to be addressed (page 8: we fix the digit movements and *take*, in the conclusion C-DSVAE uses ... to further *inject*).
ii. in equation 5, i think it would ease the reading if the KL between the aggregate posteriors is clearly defined and compared to the KL term of Eq 3 (I had to resort to the appendix for the precise definition).

(4) As future research, it would be interesting to enforce disentanglement on feedback VAEs like the one proposed in "Feedback Recurrent Autoencoder for Video Compression",  and investigate the effect on more complex temporal patterns/ compression.



**Time Spent Reviewing:**

4

---

> ### Author Response · Authors · 2021-08-10
> **We added your suggested experiments as well as clarifications on your concerns**
>
> Thanks for carefully reading our paper and giving accurate summaries. Here are some of our clarifications regarding your concerns:
>
> 1. Our implementation of contrastive estimation has generally followed the convention in the field, e.g., SimCLR[4] (for example implementation, see https://github.com/mdiephuis/SimCLR). During training, for each minibatch, an (motion/content) augmentation would be generated for each sample in the minibatch and considered as positive (motion/content) samples, which are denoted $x^+$  (see Fig. 1 and section 2.3 for examples); all other samples in the minibatch are used as negative samples w.r.t. motion/content perspective. The overhead brought by contrastive estimation is basically the same as that of MWS estimation, which requires computing aggregated posterior using all samples and computing KL between each sample’s individual posterior and the aggregated; so both estimations have complexity $O(n^2)$. You can check our code in the supplements which implemented both estimation methods.
> Our batch size is set to 128 by default (except TIMIT where bs is 256). W.r.t. the sensitivity on n, an extra experiment on TIMIT is provided here. Large batch could give good results.
> |  bs | content EER | motion EER |
> |:---:|:-----------:|:----------:|
> |  64 |    4.21%    |   30.23%   |
> | 128 |    4.07%    |   31.42%   |
> | 256 |    4.03%    |   31.81%   |
>
> 2. With regard to the MI estimation and data augmentation, we want to make several clarifications (you can also check our explanation for reviewer r1U7 in paragraph starting with “With regard to the model comparison and data augmentation, …...”):
>     * We actually have included some comparisons between contrastive estimation and all MWS estimation for Eq. 5, see Appendix F.2. In table 7 and 8, “DSVAE+all MWS est” actually means our C-DSVAE with MI terms estimated all by MWS (without the MI terms, our loss would reduce to that of DSVAE). The naming might be slightly confusing; we will make this pointC- clearer in the revision. Also, Appendix F.1 provides the ablation study for using either one augmentation.
>     * We only used data augmentation in contrastive estimation of mutual information terms, and did not use the augmented data for likelihood terms or KL terms. If we introduce augmentation to DSVAE, while DSVAE will see more data to optimize the likelihood and KL terms, we can do the same thing in C-DSVAE to further optimize the terms besides the MI terms.
>     * Per the reviewer’s request, we have performed additional experiments where we added the augmented data to the original dataset for training DSVAE, R-WAE and C-DSVAE on SM-MNIST; now all loss terms are estimated on both original and augmented data. And the results are shown in the table below. As expected, augmentation helps all methods but does not affect the relative merit (and contrastive estimation works better than MWS estimation).
> |                   | Acc    | IS    | H(y\|x) | H(y)  |
> |-------------------|--------|-------|---------|-------|
> | DSVAE+aug         | 88.89% | 6.231 | 0.199   | 2.021 |
> | C-DSVAE with all MWS est | 91.81% | 6.312 | 0.205   | 2.107 |
> | R-WAE+aug         | 95.10% | 6.962 | 0.171   | 2.135 |
> | C-DSVAE           | 97.84% | 7.163 | 0.145   | 2.176 |
>
> 3. Thanks for pointing out the typos and giving writing suggestions. We will correct and improve those in the revision.
>
> 4. We thank the reviewer for suggesting the interesting future direction.

---

### Official Review · Reviewer_r1U7 · 2021-07-18

**Rating:** 7
**Confidence:** 3

**Summary:**

This paper tackles learning disentangled representations for sequential data through contrastive estimation methods to estimate mutual information terms. Content and motion data augmentation approaches are proposed for disentangling static and dynamic factors in a sequential VAE.

**Limitations And Societal Impact:**

Seeing that improvements are from the contrastive estimation objective instead of just seeing more data-augmented samples would benefit this paper.

**Main Review:**

Overall, this paper is built on existing works but has good experimental results. In terms of originality, the ELBO formulations/modifications seem to be direct counterparts of the standard VAE to sequential data (however, theorem 1 seems to be new), and the connection between contrastive estimation and mutual information has its own history, with the CPC paper among the first to discuss the connection between contrastive estimation and mutual information. Regardless, this combination of data-augmented contrastive estimation with disentanglement seems novel and interesting.

One concern is perhaps a lack of fairness during comparison. Since the proposed method uses data augmentation during training, the comparisons to baselines should also contain data augmentation. The main difference is how the proposed method exploits the grouping of augmented sequences, while the baseline methods assume all sequences are independent. The current evaluation seems unfair because the proposed method may simply be doing better by seeing more samples, so it'd be good to see ablation experiments where baseline methods are trained on the data-augmented samples.

Since this contrastive estimation objective depends on having a large amount of negative samples, what is the change in performance across different values of n for Eq 6? How does this compare to MWS when changing the size of the minibatch? Both estimators are biased when using finite samples, so it'd be interesting to see if the bias from contrastive estimation is smaller in practice.

Is the contrastive objective with cosine similarity still an estimator for mutual information? If you plot it, does it show similar values to the MWS estimator?

Since this introduces an additional hyperparameter compared to MWS, how sensitive is the temperature tau?

Clarity: Papers on contrastive estimation using data augmentation should probably be referenced in section 2.3. The definition of positive and negative should be more clearly explained in this context. Prior works that make use of cosine similarity should also be cited. The authors should also comment on why cosine similarity is used, instead of other similarity metrics.

----

Update after author response: Thanks for the thorough response. The additional experiments sound good to me, and I have updated my score.

**Time Spent Reviewing:**

3

---

> ### Author Response · Authors · 2021-08-10
> **Clarifications on model comparison and augmentation, as well as additional experiments as suggested**
>
> Thank you for recognizing the novelty of our work. Please also note that our ELBO formulation itself is new. The direct counterpart of the standard VAE is DSVAE, which did not introduce marginalized posterior and therefore no mutual information terms were derived. Our ELBO is different and synergistically incorporates the MI terms. See section 2.2 for detailed discussion on the difference and limitations of previous formulations like DSVAE.
>
> With regard to the model comparison and data augmentation, we want to make several clarifications:
> 1. We believe our comparison is fair to the compared methods, and in fact in favor of some of them as they introduced extra loss terms that use additional weak supervision. For example, S3VAE used optimal flow maps and facial landmark detectors trained on external data to guide representation learning (see the last paragraphs of section 1 in the S3VAE paper [30]), and reordered sequence augmentation (section 4.1 in S3VAE paper). These techniques are quite dataset/domain specific, and often more expensive than our augmentations. Similarly, RWAE used weak supervision on the total number of motion classes to learn a discrete latent variable in its model parameterization (section 3.4 in RWAE paper). We discussed these aspects in the Related Work section. Our methods used no such weak supervision beyond simple augmentations.
> 2. For our objective, we only used data augmentation for the mutual information terms with contrastive estimation (and other methods did not use contrastive estimation), and we did not use the augmented data for the likelihood terms or KL terms (and other methods do have these terms). It is intuitive that all methods could benefit from seeing more data. That is, our method could benefit from using augmented data for the terms besides MI.
> 3. To verify your intuition, we have performed additional experiments where we added the augmented data to the original dataset for training DSVAE, R-WAE and C-DSVAE on SM-MNIST; now all loss terms are estimated on both original and augmented data. And the results are shown in the table below. As expected, augmentation helps all methods but does not affect the relative merit.
>
> |                   | Acc    | IS    | H(y\|x) | H(y)  |
> |-------------------|--------|-------|---------|-------|
> | DSVAE+aug         | 88.89% | 6.231 | 0.199   | 2.021 |
> | R-WAE+aug         | 95.10% | 6.962 | 0.171   | 2.135 |
> | C-DSVAE+aug       | 97.94% | 7.187 | 0.146   | 2.168 |
>
> Regarding using MWS vs. contrastive estimation for estimating the mutual information terms, we have shown in Figure 6 that the two are positively correlated: even though we optimize over the contrastive estimation, the MWS-estimated mutual information also improves. However, besides estimating mutual information, we think the additional inductive bias of contrastive estimation contributes to the superior performance, see Appendix F.2 for the comparison of the two estimates on downstream tasks. It is an interesting research question to further explore the relationship between mutual information and contrastive estimation, as probed by [35]. We are performing sensitivity analysis on the batch size and will add it to the revision. For example, in the following experiment w.r.t. different batch sizes on TIMIT, a large batch size n could give good results (both 128 and 256 are good).
>
> |  bs | content EER | motion EER |
> |:---:|:-----------:|:----------:|
> |  64 |    4.21%    |   30.23%   |
> | 128 |    4.07%    |   31.42%   |
> | 256 |    4.03%    |   31.81%   |
>
> Regarding cosine similarity: While the original derivation of contrastive estimation being MI bound (by the CPC paper) did not use cosine similarity, most later works including SimCLR, SupCon have adopted cosine similarity as a practical implementation in contrastive learning, and we followed this approach. Intuitively, cosine similarity removes the degree of freedom of the feature scale, and in general works well for high dimensional feature spaces. We will further clarify this in the future version and cite relevant papers.
>
> Regarding $\tau$: In our paper, we have set $\tau$=0.5 (it is a reasonable value given that we are using cosine similarity with range [-1, 1]). It worked well for our model and we have not tuned it. We could add some sensitivity analysis in revision.

---

### Decision · Program_Chairs · 2021-09-27

**Decision:**

Accept (Poster)

**Comment:**

This paper applies contrastive learning to improve disentanglement in sequential latent variables. The idea is based on mutual information as used in many disentangled latent variable models, and then apply contrastive learning techniques as to estimate the mutual information.

Reviewers find the proposed approach novel and interesting. Initially they raise concerns on justifications of the proposed objective and potential limitation of the experimental results. In author feedback, authors conducted extra experiments to explain the intuition of the proposed objective.

I think the idea of applying contrastive learning to disentangled sequential models is quite natural, although indeed I am not aware of other existing work that applies this idea to sequential VAEs. Experimental results indeed show advantage of the proposed approach, although it would be useful to discuss more on intuitions and analyse the key advantage of contrastive learning as compared with other mutual information based disentanglement techniques.